Manuscript prepared for Atmos. Meas. Tech.
with version 5.0 of the LaTeX class copernicus.cls.
Date: 2 June 2017

# A Climate-scale Satellite Record for Carbon Monoxide: The MOPITT Version 7 Product

Merritt N. Deeter[1], David P. Edwards[1], Gene L. Francis[1], John C. Gille[1], Sara Martínez-Alonso[1], Helen M. Worden[1], and Colm Sweeney[2]

[1]Atmospheric Chemistry Observations and Modeling Laboratory, National Center for Atmospheric Research, Boulder, CO, USA
[2]Global Monitoring Division, NOAA/ESRL, Boulder, CO, USA

*Correspondence to:* M. N. Deeter (mnd@ucar.edu)

**Abstract.**

The MOPITT ("Measurements of Pollution in the Troposphere") satellite instrument has been making observations of atmospheric carbon monoxide since 2000. Recent enhancements to the MOPITT retrieval algorithm have resulted in the release of the Version 7 (V7) product. Improvements include (1) representation of growing atmospheric concentrations of $N_2O$, (2) use of meteorological fields from the MERRA-2 reanalysis for the entire MOPITT mission (instead of MERRA), (3) use of the MODIS Collection 6 cloud mask product (instead of Collection 5), (4) a new strategy for radiance bias correction, and (5) an improved method for calibrating MOPITT's NIR radiances. Statistical comparisons of V7 validation results with corresponding V6 results are presented, using aircraft in-situ measurements as the reference. Clear improvements are demonstrated for V7 products with respect to overall retrieval biases, bias variability, and bias drift uncertainty.

## 1   Introduction

Principle sources of carbon monoxide (CO) in the troposphere include biomass burning, fossil fuel burning and the oxidation of methane and other volatile organic compounds. CO has a typical lifetime of a few months, and is destroyed by reaction with the hydroxyl radical. CO is an important precursor to ozone. Satellite measurements of CO are used in air quality forecasts (Inness et al., 2015) as well as a variety of studies of pollution sources, transport and atmospheric chemistry. MOPITT ("Measurements of Pollution in the Troposphere") is an instrument on the NASA Terra satellite designed to permit retrievals of CO vertical profiles using both thermal-infrared (TIR) and near-infrared (NIR) observations. The MOPITT instrument has been operating nearly continuously since 2000 (Drummond et al., 2010, 2016), resulting in a homogeneous long-term data record well

suited for a variety of applications, including trend analyses (Worden et al., 2013; Strode et al., 2016; Xia et al., 2016). MOPITT retrieval products have improved continuously as the result of accumulated knowledge regarding the instrument, forward modeling methods, and geophysical variables (Worden et al., 2014).

MOPITT CO retrieval products are available in three variants: TIR-only, NIR-only, and the multispectral TIR-NIR product (Deeter et al., 2013). These three products exhibit contrasting retrieval sensitivity and error characteristics. The TIR-NIR product offers the greatest vertical resolution, and particularly the greatest sensitivity to CO in the lower troposphere. However, this product also exhibits relatively large random retrieval errors and bias drift. Moreover, the main benefits of this product are only evident in daytime MOPITT observations over land, due to limitations of the NIR radiances. The TIR-only product offers the highest temporal stability and generally similar performance in variable observing situations (day and night, land and ocean). The NIR-only product is primarily suited for the analysis of CO total columns and is produced solely for daytime observations over land.

## 2 V7 Algorithm Features

Refinements to the MOPITT retrieval algorithm and data processing systems are developed and implemented in new products for a variety of reasons. For example, the MOPITT retrieval algorithm incorporates a fast radiative transfer model which is periodically updated as new sources of model error are identified and corrections are incorporated. Reprocessing is also necessary as auxiliary datasets used in MOPITT retrieval processing (such as meteorological reanalyses and MODIS cloud mask files) are themselves refined and reprocessed. Finally, processing parameters in the retrieval algorithm are also revised as new validation datasets become available, resulting in improved understanding of MOPITT retrieval errors and long-term bias drift. Specific improvements incorporated into the MOPITT Version 7 product are described in detail below.

### 2.1 Radiative Transfer Modeling

The operational radiative transfer model on which the MOPITT retrieval algorithm is based, known as MOPFAS (Edwards et al., 1999), has been updated for V7. Whereas previous versions of MOPFAS assumed constant concentrations of $N_2O$, the model now accounts for the weak but steady growth of atmospheric $N_2O$ concentrations over the MOPITT mission. Overlap of $N_2O$ spectral lines with the CO TIR passband near 4.7 $\mu$m (Pan et al., 1995) suggests that slowly increasing atmospheric concentrations of $N_2O$ (Sweeney et al., 2015) could produce a time-dependent bias in model-calculated radiances for previous MOPITT products, possibly leading to retrieval bias drift.

Potential retrieval biases due to increasing atmospheric concentrations of $N_2O$ were simulated using the radiative transfer model MOPABS (Edwards et al., 1999); this model is computationally

less efficient than MOPFAS, but is more accurate and more flexible. Over a representative atmospheric ensemble (Deeter et al., 2010), forward model radiances were calculated using two versions of MOPABS. In one version, $N_2O$ was fixed at the V6 operational value; in the second version it was elevated by 3% at all levels. (With respect to the observed $N_2O$ growth rate described in the following paragraph, a 3% increase roughly corresponds to the accumulated growth over 12 years.) These two sets of model-calculated radiances were then passed to the V6 retrieval algorithm in order to estimate the CO retrieval bias due to increasing $N_2O$. Simulated biases due to the 3% increase in $N_2O$ are shown in Figure 1. While the predicted biases exhibit significant variability, increasing $N_2O$ typically results in negative retrieval biases in the lower troposphere and smaller positive biases in the upper troposphere. Simulated $N_2O$-related biases are typically a few ppbv, but can reach 10-15 ppbv. These simulations indicate that, if not represented in the radiative transfer model, increasing $N_2O$ may produce retrieval bias drift up to approximately 1 ppbv per year, but with opposite effects in the lower and upper troposphere. This is a potentially significant drift, particularly for CO climate applications.

For V7, the time-dependent $N_2O$ concentration estimate used in the forward model is derived from global monthly means provided by the NOAA/ESRL halocarbons program. The $N_2O$ forward model calculation makes use of a fixed Global Monitoring Division dataset covering the period from September 1977 through August 2015 (ftp://ftp.cmdl.noaa.gov/hats/n2o/combined/HATS_global_N2O.txt). These data are well described by a linear increase in $N_2O$ concentrations from 300 ppb to 328 ppb over that time, a growth rate of approximately 0.7 ppb/yr. The forward model calculation used for V7 processing performs a linear fit to these data, then interpolates (or extrapolates) the fit to the mid-month date for the specified monthly forward model. For dates in 2016, the resulting $N_2O$ values are (on average) about 8% larger than the constant value of 303 ppb assumed for earlier MOPITT data versions. The operational radiative transfer model has also been updated with the HITRAN 2012 spectral database (Rothman et al., 2013).

## 2.2 Meteorological Fields

For each retrieved CO vertical profile at a particular location, the MOPITT retrieval algorithm requires temperature and water vapor profiles as well as a priori surface temperature values (Deeter et al., 2003). For V6 processing, these meteorological data were derived from the NASA MERRA ("Modern-Era Retrospective Analysis for Research and Applications") reanalysis product (https://gmao.gsfc.nasa.gov/reanalysis/MERRA/). MERRA production ceased in 2016. For all V7 products, meteorological profiles are extracted from the more recently released MERRA-2 product; this version is continuously available for the entire MOPITT data record. As described here (https://gmao.gsfc.nasa.gov/reanalysis/MERRA-2/), the MERRA-2 assimilation system exploits a wider range of modern hyperspectral radiance and microwave observations, along with Global Po-

sitioning System (GPS) occultation datasets. It also uses NASA ozone observations (e.g., from the MLS and OMI instruments) starting in 2004. Advances in both the GEOS-5 model and the Grid-point Statistical Interpolation (GSI) assimilation system are also included in MERRA-2. Generally

higher quality retrieval results for MOPITT V7 processing are expected using MERRA-2 due to the assimilation of more satellite datasets and other improvements.

## 2.3 Cloud Detection

Only MOPITT observations of clear-sky scenes are passed to the retrieval algorithm. The clear/cloudy determination is based both on MOPITT's thermal-channel radiances and the MODIS

cloud mask (Deeter, 2011). Since about 2010, electronic crosstalk affecting MODIS thermal-channel Bands 29 to 31 (as explained in http://modis-atmos.gsfc.nasa.gov/validation_35.html) has resulted in a false trend suggesting increasing cloudiness. This effect is most pronounced for tropical nighttime scenes over the ocean. This issue affects MODIS products from both Collection 5 (used in MOPITT V5 and V6 processing until February, 2016) and Collection 6 (used in MOPITT V5 and V6 products

since March, 2016).

For the cloud detection algorithm used for MOPITT V7 processing, two changes have been made to mitigate issues associated with the quality of the MODIS cloud mask files. First, MODIS Collection 6 cloud mask files are used consistently for processing the entire MOPITT mission. (Characteristics of the Collection 6 cloud mask files are described in http://modis-

110 atmos.gsfc.nasa.gov/Webinar2014/MODIS_C6_MOD35_Ackerman.pdf.) Second, the Level 2 Cloud Description diagnostic now includes a new possible value ("6") to identify ocean scenes (both night and day) where the MODIS cloud mask-based tests indicate that the area was cloudy (with the exception of scenes with low clouds only) but the test based on MOPITT's thermal-channel radiances determines that the area was clear. Such scenes were previously discarded by the cloud detection

algorithm but are now retained. Compared to earlier MOPITT products, the addition of this new class of observations may significantly increase the number of MOPITT retrievals in a given scene. The other possible Cloud Description diagnostic index values (1-5) retain their original meanings, as defined in the V5 User's Guide (Deeter, 2011).

The effect of the changes in the V7 cloud detection algorithm are illustrated in Figure 2. The

120 two panels in the figure compare the total number of clear-sky MOPITT observations (regardless of the Cloud Description diagnostic values) produced by the V6 and V7 cloud detection schemes for daytime and nighttime scenes in the Tropics (between 30°S and 30°N), from 2008 (two years before the MODIS crosstalk issue first became evident) through 2015. Whereas the number of clear-sky scenes in the Tropics decreased sharply in recent years in the V6 MOPITT product (particularly for

nighttime scenes), no such trend is apparent for the V7 product.

## 2.4 Radiance Bias Correction

The MOPITT Level 2 processor exploits a set of fixed radiance-bias correction factors to compensate for relative biases between simulated radiances calculated by the operational radiative transfer model and actual calibrated Level 1 radiances. Without some form of compensation, radiance biases produce biases in the retrieved CO profiles. Radiance-bias correction factors counteract a variety of potential bias sources including errors in instrumental specifications, forward model errors, spectroscopy errors, and geophysical errors.

New strategies were developed for deriving radiance-bias correction factors for V7 products. For the TIR radiances (Channels 5 and 7), radiance-bias scaling factors were determined by minimizing observed retrieval biases at 400 and 800 hPa using in-situ CO profiles from the HIPPO (HIAPER Pole to Pole Observations) field campaign (Deeter et al., 2013; Martínez-Alonso et al., 2014; Deeter et al., 2014). To the extent that the HIPPO campaign produced a near-global set of in-situ CO profiles (i.e., over a wide latitudinal range spanning both the Northern and Southern Hemispheres), this strategy yields globally-minimized retrieval biases. For the NIR radiances (Channel 6), radiance-bias scaling factors were determined by minimizing NIR-only retrieval biases as determined using the NOAA aircraft profile set described in Section 3.1. (The HIPPO dataset primarily represents oceanic scenes and was therefore not useful for optimizing the NIR radiance-bias scaling factors.) Previous MOPITT products did not apply radiance-bias scaling factors to compensate for NIR radiance biases.

## 2.5 Calibration

Calibration of MOPITT's NIR channel (i.e., Channel 6) relies on a two-point calibration scheme involving both cold-calibration ("cold-cal") events and hot-calibration ("hot-cal") events. Calibration information from the cold-cal and hot-cal events are used to derive gain and offset values for calibrating the earth-view radiances (Deeter et al., 2002). Cold-cals occur many times per day, while hot-cals are performed only about once per year (Drummond et al., 2010). Ideally, NIR channels are calibrated with gain and offset values determined by interpolating the information from "bracketing" hot-cals occurring both before and after the time of observation. While this method is feasible in retrospective processing mode (i.e., processing data from previous years), it is not possible in forward processing mode (i.e., when processing recently acquired observations). Thus, in forward processing mode, only information from the most recent hot-cal is used to calibrate MOPITT's NIR radiances. Recent comparisons of V5 and V6 NIR-only retrieval products generated in retrospective and forward processing modes have revealed significant differences (up to about 20%) in total column results, with the retrospectively processed data in better agreement with daytime/land TIR-only total column values.

Therefore, because of the lower quality of MOPITT products processed in forward processing mode, all V7 products generated in this manner will be clearly identified as "beta" products (i.e.,

"beta" will clearly appear in the filename for all such products). These products will be reprocessed and replaced by standard archival files following the next hot-cal. Typically, this will occur no more than about 14 months from the time of a particular observation (depending on the date of the most recent hot-cal). For example, final V7 products for observations made between March, 2016 and March, 2017 (months during which hot-cals occurred) should become available to users around May, 2017.

In principle, this new strategy should benefit not only the NIR-only and TIR-NIR products, but also the TIR-only products. Calibration of the TIR Pressure Modulation Cell (PMC) radiances (Channel 7) involves annual measurements of mean cell pressure obtained by measuring the PMC resonant frequency (Drummond et al., 2010); these cell pressure measurements occur in conjunction with the annual hot-cals. Like the NIR radiances, Channel 7 radiances calibrated with bracketing mean cell pressures are considered more reliable than radiances based only on the most recent previous cell pressure measurement.

V7 beta products should not be used for examining long-term records of CO although these products should still be useful for some applications. Differences in quality between beta and archival products are expected to be greatest for the NIR-only products, but could be significant for the TIR-only and TIR-NIR products as well.

## 3    Validation Results

Below, retrieval validation results for V7 products are compared to corresponding V6 results (Deeter et al., 2014). Validation results are based on statistical comparisons of MOPITT retrieval products (CO volume mixing ratio profiles and total columns) with in-situ profiles measured from aircraft. For this purpose, in-situ measurements are assumed to be exact and representative of an extended region around the sampling location. A collocation radius of 50 km was employed for the NOAA profiles whereas a value of 200 km was used for the HIPPO profiles. The larger acceptance radius for the HIPPO profiles was selected due to the weaker CO gradients expected over the remote ocean (far from CO source regions). In both cases, a maximum of 12 hours was allowed between the time of the MOPITT observation and acquisition of the in-situ data.

Because of the coarse vertical resolution of the radiance weighting functions (or "Jacobians") and the underconstrained nature of the retrieval process, retrieval products obtained with optimal estimation-based retrieval algorithms are constrained by a priori information as well as the measurements (Pan et al., 1998; Rodgers, 2000). A priori information is represented by (1) an a priori profile $x_a$ and (2) an a priori covariance matrix, which determines the strength of the a priori constraint. The relationship between the true profile $x_{true}$, $x_a$, and retrieved profile $x_{rtv}$ is expressed by the equation

$$x_{rtv} = x_a + \mathbf{A}(x_{true} - x_a) \tag{1}$$

where $\mathbf{A}$ is the retrieval averaging kernel matrix. The vector quantities $x_{true}$, $x_a$ and $x_{rtv}$ are expressed in terms of $log(VMR)$ rather than $VMR$ itself; this strategy is justified by observations of CO variability in the troposphere (Deeter et al., 2007). $\mathbf{A}$ quantifies the sensitivity of the retrieved profile to the true profile and is provided as a diagnostic for each retrieval in all MOPITT products. $\mathbf{A}$ depends on the weighting functions, a priori covariance matrix and instrument error covariance matrix.

Validation results are presented in Sections 3.1 and 3.2 as scatterplots. However, in contrast to previous publications of MOPITT validation results, where MOPITT retrieved VMR values were plotted directly against simulated VMR retrievals $x_{sim}$ (calculated according to the ~~RHS~~ right-hand side of Eq. 1, with $x_{true}$ based on the in-situ profile), scatterplots presented in Section 3.1 and 3.2 instead present validation results in terms of the difference between the retrieved and a priori VMR values, i.e.,

$$\Delta log(VMR) = x_{rtv} - x_a \tag{2}$$

Correlation coefficients $r$ calculated from values of $\Delta log(VMR)$ reflect the correlation due to the measurement, whereas $x_{rtv}$-based correlation coefficients may be strongly influenced by variability of the a priori. For example, as the averaging kernels tend toward 0 (i.e., as retrieval sensitivity decreases), values of $r$ calculated from values of $\Delta log(VMR)$ will tend toward 0 (since both observed and simulated values of $\Delta log(VMR)$ will tend toward 0) while $x_{rtv}$-based correlation coefficients will tend toward unity (since observed and simulated values of $x_{rtv}$ will be identically affected by a priori variability). Thus, correlation coefficients calculated in terms of $\Delta log(VMR)$ are preferred for quantifying retrieval performance. For each overpass of a validation site on a particular date (i.e., a single MOPITT scene), mean retrieved values of $\Delta log(VMR)$ are shown on the y-axis (along with error bars for the standard deviation) whereas simulated values based on the in-situ measurements (i.e., $\mathbf{A}(x_{true} - x_a)$) are plotted on the x-axis. For CO total column, V6 and V7 validation results are presented in terms of retrieved values, with no subtraction of the a priori influence. This choice was made because (1) retrieved total column values are inherently less affected by the a priori than levels in the retrieved profile and (2) a priori total column values, which would be necessary to employ the same strategy used for the profile levels, were not included in V6 Level 2 product files (though they are included as diagnostics in V7 Level 2 files).

### 3.1 NOAA Flask Samples

Historically, in-situ measurements of CO concentrations acquired through NOAA's aircraft flask sampling program have served as the foundation for MOPITT validation efforts (Emmons et al., 2004, 2009; Deeter et al., 2014). Flask samples obtained from aircraft are processed by the Global Monitoring Division of NOAA's Earth System Research Laboratory (ESRL) in Boulder, Colorado. NOAA stations used for MOPITT validation are primarily located in North America (Sweeney

et al., 2015). Aircraft profiles acquired from the start of the MOPITT mission through February, 2016 were exploited to validate the MOPITT V7 product. Flask samples are typically acquired from near the surface up to about 350-400 hPa. Typical in-situ profiles are derived from approximately 12 0.7 L flasks filled to 40 PSIA. In order to obtain a complete validation profile for comparison with MOPITT retrievals, each in-situ profile is extended vertically above the highest-altitude in-situ measurement using the CAM-chem chemical transport model (Lamarque et al., 2012) and then resampled to the standard pressure grid used for the MOPITT operational radiative transfer model (Martínez-Alonso et al., 2014). The entire database of NOAA aircraft profiles (http://www.esrl.noaa.gov/gmd/ccgg/aircraft/index.html) acquired during the MOPITT mission currently includes more than 5000 CO profiles.

Validation results based on the NOAA flask samples for the V6 and V7 TIR-only products are presented in Figures 3 and 4. For simplicity, VMR validation results are only shown for alternating retrieval levels (i.e., 200, 400, 600, 800 hPa, and surface) and for CO total column. Each panel lists the overall bias, standard deviation, and correlation coefficient. Fractional biases are calculated from retrieved and simulated $log(VMR)$ values by exploiting the relation

$$\delta(VMR)/VMR \approx \delta(ln(VMR))$$
$$= \delta(log(VMR))/log(e)$$
$$= (log(VMR_{rtv}) - log(VMR_{sim}))/log(e)$$
$$= (x_{rtv} - x_{sim})/log(e) \tag{3}$$

Fractional standard deviation values are calculated similarly. The factor $log(e)$ is the result of expressing $log(VMR)$ values using base-ten logarithms rather than natural logarithms and has no physical significance. Percentage bias and standard deviation values are obtained by simply multiplying the fractional bias and standard deviation values by 100. Validation results based on the NOAA flask samples for the V6 and V7 NIR-only products are presented in Figures 5 and 6, whereas results for the V6 and V7 TIR-NIR products are presented in Figures 7 and 8. Validation statistics for V6 and V7 products are also summarized in Table 1.

Statistical significance was investigated for the data in each scatterplot presented in Figures 3-8 using the t-test for the correlation coefficient. With the exception of the V6 TIR-NIR results at 200 hPa (which yielded a correlation coefficient of 0.03 and a p-value of 0.43), all of the correlations for V6 and V7 products were found to "highly significant" ($p < 0.05$).

## 3.2 HIPPO Measurements

The "HIAPER Pole to Pole Observations" (HIPPO) campaign included five phases of operations between 2009 and 2011 (Wofsy et al., 2011). The extensive coverage of the HIPPO flights makes this dataset useful for analyzing the geographical dependence of retrieval biases (Deeter et al., 2013). In-situ measurements of atmospheric composition were performed using the QCLS ("Quantum-

260 Cascade Laser Spectrometer") instrument (Santoni et al., 2014) from approximately 67°S to 80°N mostly over the Pacific Ocean. In-situ profiles produced with the HIPPO measurements were extended vertically with the CAM-chem climatology, in the same manner as described in Section 3.1. A total of 567 in-situ CO profiles acquired during the five phases of HIPPO were used for MOPITT validation. HIPPO flights were performed during January, 2009 (Phase 1), October/November, 2009

(Phase 2), March/April, 2010 (Phase 3), June/July, 2011 (Phase 4), and August/September, 2011 (Phase 5). Each of the profiles used for validation include measurements made at a minimum pressure of 400 hPa or less; 141 HIPPO profiles actually reached 200 hPa or less. In addition, all profiles used for validation reached a maximum pressure of at least 800 hPa, and included vertical gaps (lacking in-situ data) no larger than 200 hPa.

Since MOPITT NIR observations can only be exploited in daytime scenes over land, the HIPPO profiles are used here only to evaluate the V7 TIR-only retrieval products. Validation results based on HIPPO CO profiles for the V6 and V7 TIR-only products are presented in Figures 9 and 10. HIPPO validation statistics are also summarized in Table 2. Correlations of the scatterplots presented in Figures 9 and 10 were all found to be highly significant based on the t-test results.

**4 Analysis**

Retrieval improvements associated with the new algorithm features described in Section 2 should be evident in comparisons of V6 and V7 validation statistics. However, because of the sparseness of aircraft in-situ measurements at high altitudes (e.g., pressures less than 350 hPa), particularly for the NOAA dataset, statistical comparisons of V6 and V7 upper-tropospheric CO products are

280 less significant than comparisons of results for the lower troposphere. For example, for retrievals of CO at 200 hPa, the sections of the NOAA validation profiles in the upper troposphere and lower stratosphere are heavily based on the CAM-chem climatology (as described in Section 3.1), and validation results will likely be less reliable than for lower levels. Methods and datasets useful for validating MOPITT upper-tropospheric CO concentrations were reported in Martínez-Alonso et al.

(2014).

**4.1 TIR-only**

For the validation results shown in Figures 3 and 4, biases for the V6 and V7 TIR-only products are on the order of a few percent or less. TIR-only biases are more clearly improved for the HIPPO profiles shown in Figures 9 and 10, where, for example, the bias at 600 hPa improved from -4.7%

to -0.9%. However, for both the NOAA and HIPPO profiles, the standard deviation statistic is significantly improved in the V7 TIR-only products. This statistic represents the variability of the single-scene biases calculated over all overpasses. For the retrieved CO total column, for example, the standard deviation derived from the NOAA validation sites is reduced from $0.17 \times 10^{18}$

~~mol~~molec/cm$^2$ to $0.13 \times 10^{18}$ ~~mol~~molec/cm$^2$. Smaller standard deviations and larger correlation
coefficients are also observed for the VMR validation results. For example, for the HIPPO results
at 600 hPa, the standard deviation decreased from 10.0% to 8.1% and the correlation coefficient
increased from 0.56 to 0.65. Relatively low correlation coefficients at 200 hPa for both the V6 and
V7 TIR-only results presented in Figures 3 and 4 are likely the result of both (1) errors in the in-situ
profiles associated with the lack of actual in-situ data at high altitudes (as discussed above) and (2)
weaker CO geophysical variability in comparison to the variability at lower altitudes.

There is no apparent reason why random retrieval errors would be significantly different for V7
products than for V6, since no significant change was implemented in the method for calibrating
the V7 TIR radiances or calculating radiance uncertainties. The observed reduction in standard de-
viations for V7 TIR-only products thus implies the suppression of a bias source (or sources) which
varies either temporally, geographically, or both. This improvement is likely associated with ei-
ther changes made to the MOPITT radiative transfer model, the source of meteorological fields
(MERRA-2), or improvements in cloud detection.

### 4.2 NIR-only

Comparisons of V6 and V7 NIR-only validation results for the NOAA profiles shown in Figures 5
and 6 also indicate significant improvements for V7. Overall VMR biases are reduced substantially
at all levels, from about 7% to -1.4%, likely from the use of radiance correction for the V7 NIR-
only products, as described in Section 2.4. The bias in total column is reduced from $0.15 \times 10^{18}$
~~mol~~molec/cm$^2$ to $-0.01 \times 10^{18}$ ~~mol~~molec/cm$^2$. NIR-only standard deviations are also improved.
For the retrieved CO total column, for example, the standard deviation is reduced from $0.18 \times 10^{18}$
~~mol~~molec/cm$^2$ to $0.12 \times 10^{18}$ ~~mol~~molec/cm$^2$. This could be the result of improvements in the
MOPITT radiative transfer model, improved meteorological fields (MERRA-2), or cloud detection.
The revised NIR calibration scheme could also significantly improve V7 validation results, although
this would only be relevant for observations made after March 17, 2012 (i.e., the period for which
V6 NIR-only products were processed in forward processing mode).

### 4.3 TIR-NIR

Finally, validation results for V7 TIR-NIR retrievals also exhibit clear improvements compared to
V6, as indicated in Figures 7 and 8. Whereas biases in the lower troposphere for V6 vary from
-3.7% to 8.3%, V7 biases fall in the narrower range from -3.4 to 2.8%. The overall bias in total
column is reduced from $0.08 \times 10^{18}$ ~~mol~~molec/cm$^2$ to $0.03 \times 10^{18}$ ~~mol~~molec/cm$^2$. Standard devi-
ations also decrease substantially while correlation coefficients are generally greater. The greatest
improvement for V7 TIR-NIR products is found for the surface-level retrieval, for which the bias
decreases from 8.3 to 2.8%, the standard deviation decreases from 18 to 11%, and the correlation
coefficient increases from 0.29 to 0.50.

### 4.4 Long-term Stability

As the longest satellite record of atmospheric CO, already 17 years, MOPITT data are increasingly used for climate applications (e.g., Worden et al. (2013); Strode et al. (2016)). However, since long-term retrieval bias drift can mimic the effect of a trend in atmospheric CO concentrations, this effect should be explicitly considered in all long-term analyses of MOPITT data. Bias drift in MOPITT products could result from a variety of sources, including long-term instrumental degradation, long-

term changes in the quality of datasets used in MOPITT processing (e.g., meteorological fields), and geophysical effects which are not represented in the MOPITT radiative transfer model (such as the variability of trace gases other than CO and water vapor). Characterizing retrieval bias drift (and its uncertainty) is thus critical for exploiting MOPITT's long record. The continuity of NOAA's aircraft flask sampling program over the MOPITT mission enables such an analysis, at least with respect to

North America. Bias drift in other geographical regions could, in principle, be different.

     Timeseries of retrieval bias ($x_{rtv}$ - $x_{sim}$) for the V6 and V7 TIR-only, NIR-only, and TIR-NIR products are presented in Figures 11-16. Each panel also lists the bias drift (in %/yr) and associated uncertainty, as determined from a least-squares fit. Bias drift values are also listed in Table 1. As shown in Figures 11 and 12, both the V6 and V7 TIR-only products exhibit negative long-term drift

in the lower troposphere and positive drift values in the upper troposphere. And, for both V6 and V7, long-term drift for the CO total column ($0.002\pm0.001 \times 10^{18}$ ~~mol~~molec/cm$^2$/yr) is nearly negligible. However, bias drift is reduced for V7 at 600 and 800 hPa. Drift uncertainty values are also generally smaller for V7 than for V6, which appears to be related to the smaller standard deviation values for V7 reported in Section 4.1. The statistical significance of the timeseries least-squares fits, based on

the t-test on the slope, was also investigated. For the TIR-only products, fitted least-squares lines were all "highly significant" ($p < 0.05$) except for the V6 600 hPa VMR product ($p = 0.57$), the V6 total columns ($p = 0.24$) and V7 total columns ($p = 0.10$).

     For the NIR-only results shown in Figures 13 and 14, bias drift values for V6 and V7 are similarly small in magnitude, but with opposite signs. For example, whereas the bias drift at the surface

for V6 is $0.29\pm0.12$ %/yr, the corresponding drift for V7 is $-0.25\pm0.09$ %/yr. For both V6 and V7 NIR-only products, fitted least-squares lines were all highly significant. Qualitatively, the bias drift values for the V6 and V7 TIR-NIR products shown in Figures 15 and 16 behave similarly to the TIR-only drift values: positive bias drift is observed in the upper troposphere, negative drift is observed in the lower troposphere, and the total column bias drift is nearly negligible. Overall, bias

drift values for V7 TIR-NIR retrievals are slightly smaller relative to V6, except at the surface. Bias drift uncertainty values are also smaller for V7. Fitted least-squares lines for V6 and V7 TIR-NIR bias timeseries were highly significant for all VMR products, but not for total column ($p = 0.38$ for V6 and $p = 0.59$ for V7).

## 4.5 Geographical Dependence of Biases

The geographical variability of MOPITT retrieval biases was first studied for the V5 product using the HIPPO dataset (Deeter et al., 2013). Because HIPPO flights were primarily performed over the ocean, these data are only useful for analyzing the geographical variability of biases in TIR-only products. The physical source of latitude-dependent biases has not been identified, but is unlikely to be related to the MOPITT instrument. A partial list of possible causes includes (1) geographically variable biases in the meteorological data used in the retrieval algorithm, (2) geographical variability of trace gases assumed to be fixed in the radiative transfer model (e.g., $N_2O$ and methane), and (3) radiative effects due to clouds.

MOPITT V6 and V7 TIR-only retrieval biases calculated with the HIPPO in-situ profiles are plotted versus latitude in Figures 17 and 18. Large black diamonds and error bars in each panel indicate bias statistics (mean and standard deviation) representing each 30 degree-wide latitudinal zone; these results are also summarized in Table 3. Overall, the geographical dependence of the V7 biases is slightly reduced compared to V6. For example, V6 TIR-only validation results indicate a significant negative bias at 800 hPa in the northern Tropics (between the Equator and 30°N), which is reduced in the V7 results. Moreover, a comparison of the lengths of the error bars in all panels of Figures 17 and 18 indicates that bias variability within each latitudinal zone is also generally smaller for V7 than for V6.

## 5 Conclusions

Algorithm features introduced in the V7 product have particularly improved the temporal consistency of the retrievals. For example, the effects of gradually increasing concentrations of $N_2O$ on MOPITT's thermal-channel radiances, which could contribute to bias drift, are now explicitly represented in the operational radiative transfer model. Recent degradation in the quality of the MODIS cloud mask, primarily in tropical oceanic scenes, was addressed by forcing the V7 cloud detection algorithm to pass all scenes where MOPITT thermal channel radiances indicate clear skies, even if the MODIS cloud mask indicates cloudiness. This change, coupled with the transition to the MODIS Collection 6 cloud mask, yields greatly improved long-term stability in the fraction of MOPITT observations passed to the retrieval algorithm. Use of the MERRA-2 reanalysis instead of the older MERRA product presumably provides higher-quality temperature and water vapor profiles needed by the retrieval algorithm. A new calibration strategy was implemented for V7 which completely relies on the interpolation of information from calibration events both before and after a particular observation. This strategy is beneficial for NIR radiance calibration, but creates a delay in the delivery of all standard archival products until the following annual hot-calibration event occurs. Users who wish to access V7 retrieval products based on preliminary calibration information will therefore have access to "beta" products, typically with a latency of a few months.

Comparisons of V6 and V7 validation results indicate clear improvements for V7. In contrast to V6, overall biases for V7 are a few percent or less at all levels for the TIR-only, NIR-only, and TIR-NIR products. Bias variability is significantly improved for V7 also, as indicated by comparisons of scatterplot standard deviations and correlation coefficients. For TIR-NIR surface-level retrievals, for example, validation statistics are significantly improved for overall bias (8.3% vs. 2.8%), bias variability (18% vs. 11%) and correlation coefficient (0.29 vs. 0.50). With respect to bias drift, improvements for V7 are modest, although V7 bias drift uncertainty values are clearly smaller than for V6. For trend analyses, this suggests that bias drift in V7 products can be corrected to a higher degree of accuracy than for V6.

*Acknowledgements.* We thank Florian Nichitiu, at the University of Toronto, for identifying the MODIS cloud mask problem in the MOPITT Version 6 product. CO measurements acquired on aircraft during the HIPPO campaign were produced by Steve Wofsy, Bruce Daube and Jasna Pittman. $N_2O$ data were obtained from the "Halocarbons and other Atmospheric Trace Gases" program within NOAA's Global Monitoring Division. Debbie Mao and Dan Ziskin provided critical software engineering and data management support. The National Center for Atmospheric Research (NCAR) is sponsored by the National Science Foundation. The NCAR MOPITT project is supported by the National Aeronautics and Space Administration (NASA) Earth Observing System (EOS) Program.

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

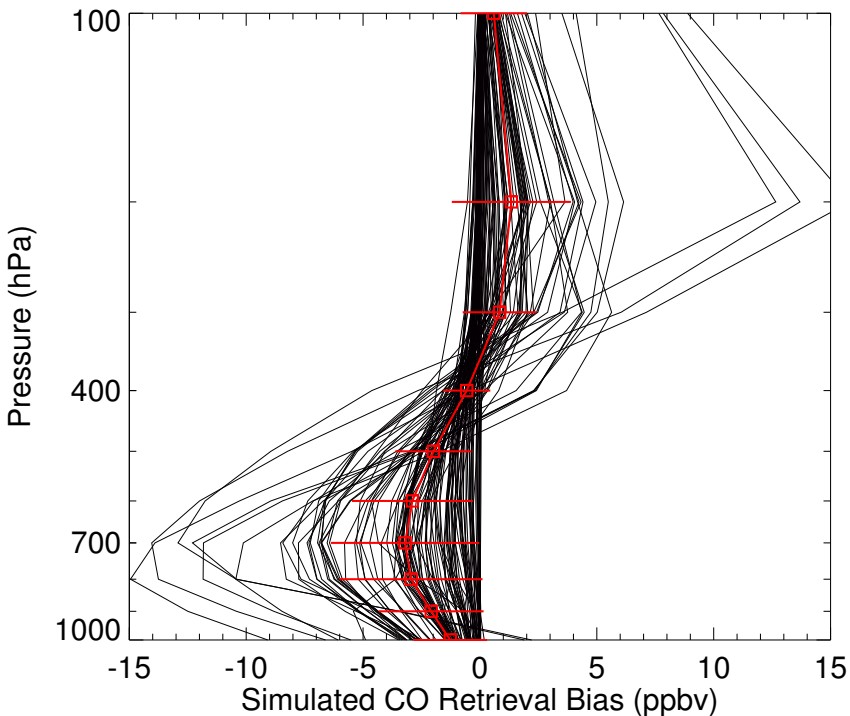

**Fig. 1.** Simulated retrieval biases due to 3% increase in atmospheric $N_2O$ concentrations.

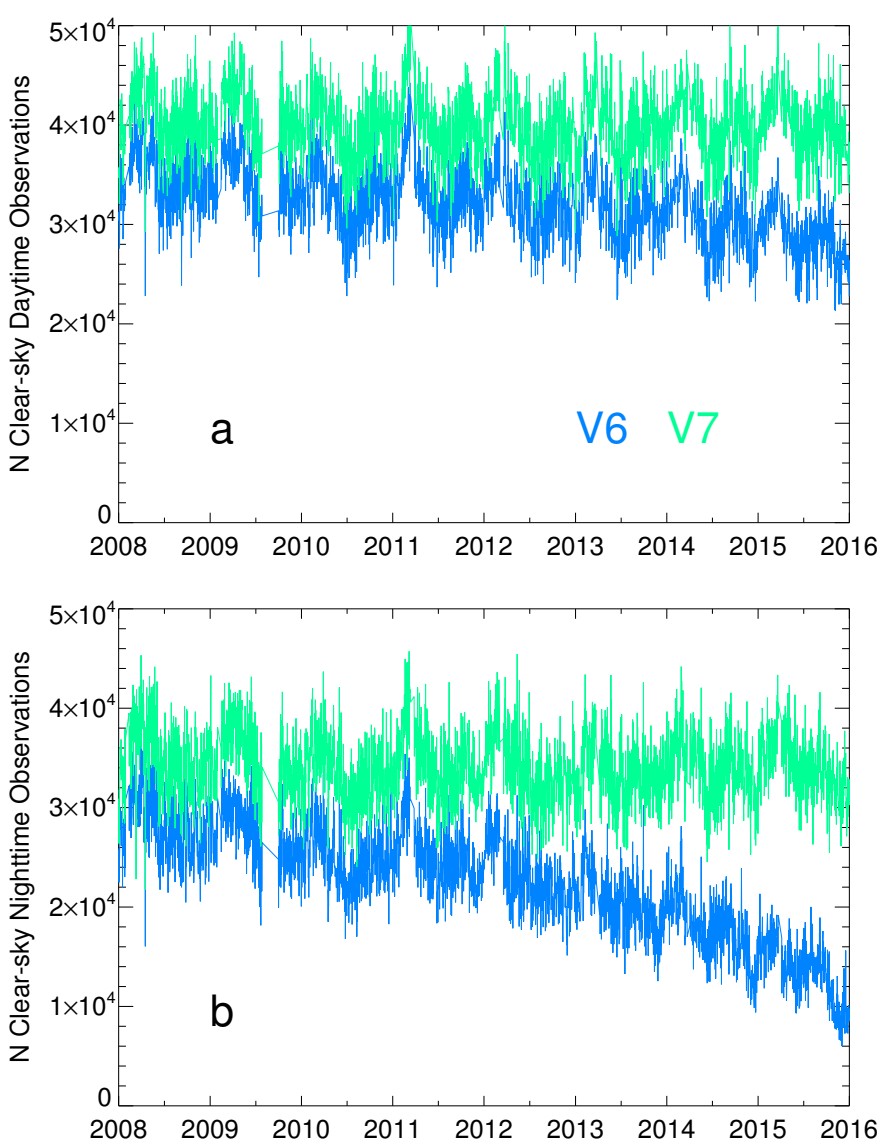

**Fig. 2.** Timeseries comparisons of (a) daytime and (b) nighttime daily number of clear-sky MOPITT observations over the ocean between 30°S and 30°N for the V6 and V7 products.

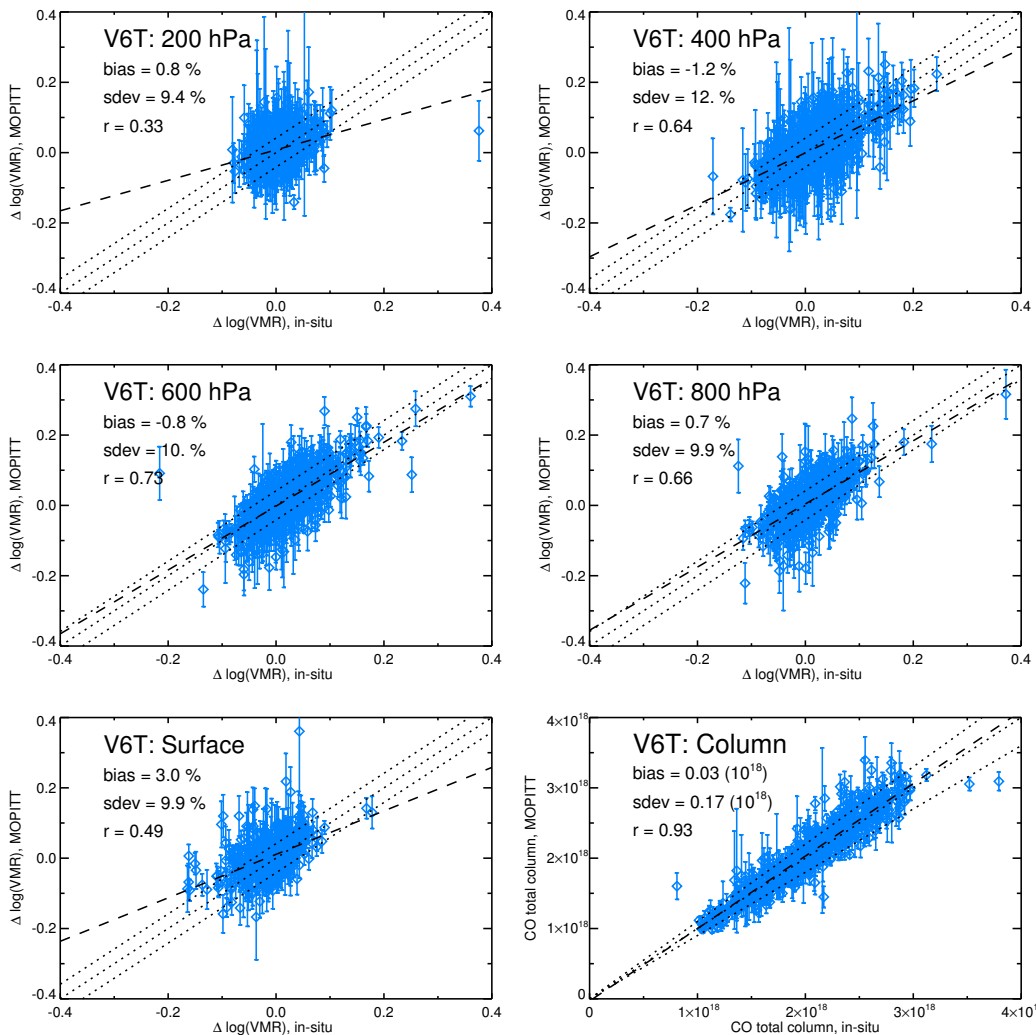

**Fig. 3.** V6 TIR-only validation results based on the NOAA flask measurements. As discussed in Section 3.1, VMR validation results are presented in terms of log(VMR), after subtracting a priori values. Dotted lines represent biases of -10, 0, and 10%. Dashed lines are least-squares best fits. ~~Bias~~ CO total column values as well as bias and standard deviation statistics for the total column are in units of ~~mol~~molec/cm$^2$.

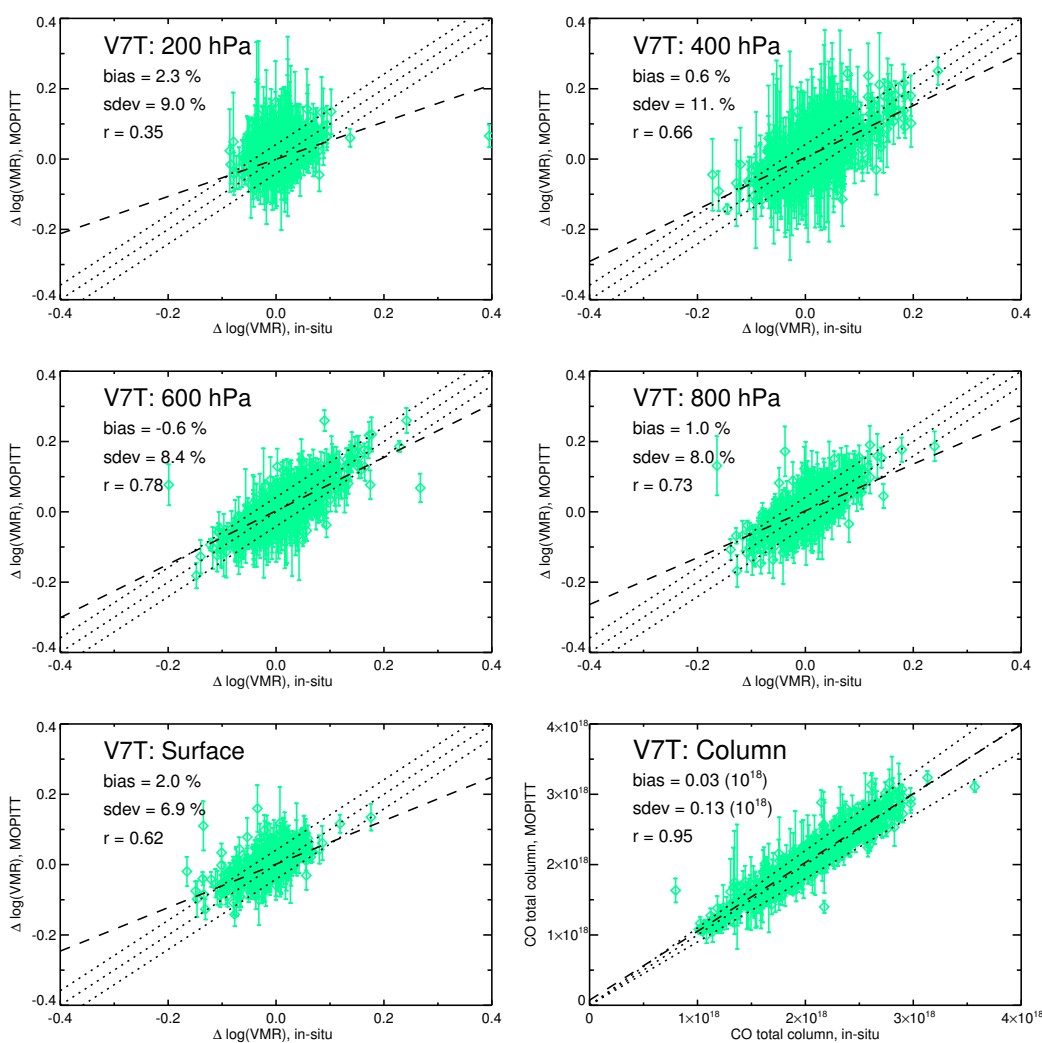

**Fig. 4.** V7 TIR-only validation results based on the NOAA flask measurements. See caption to Figure 3.

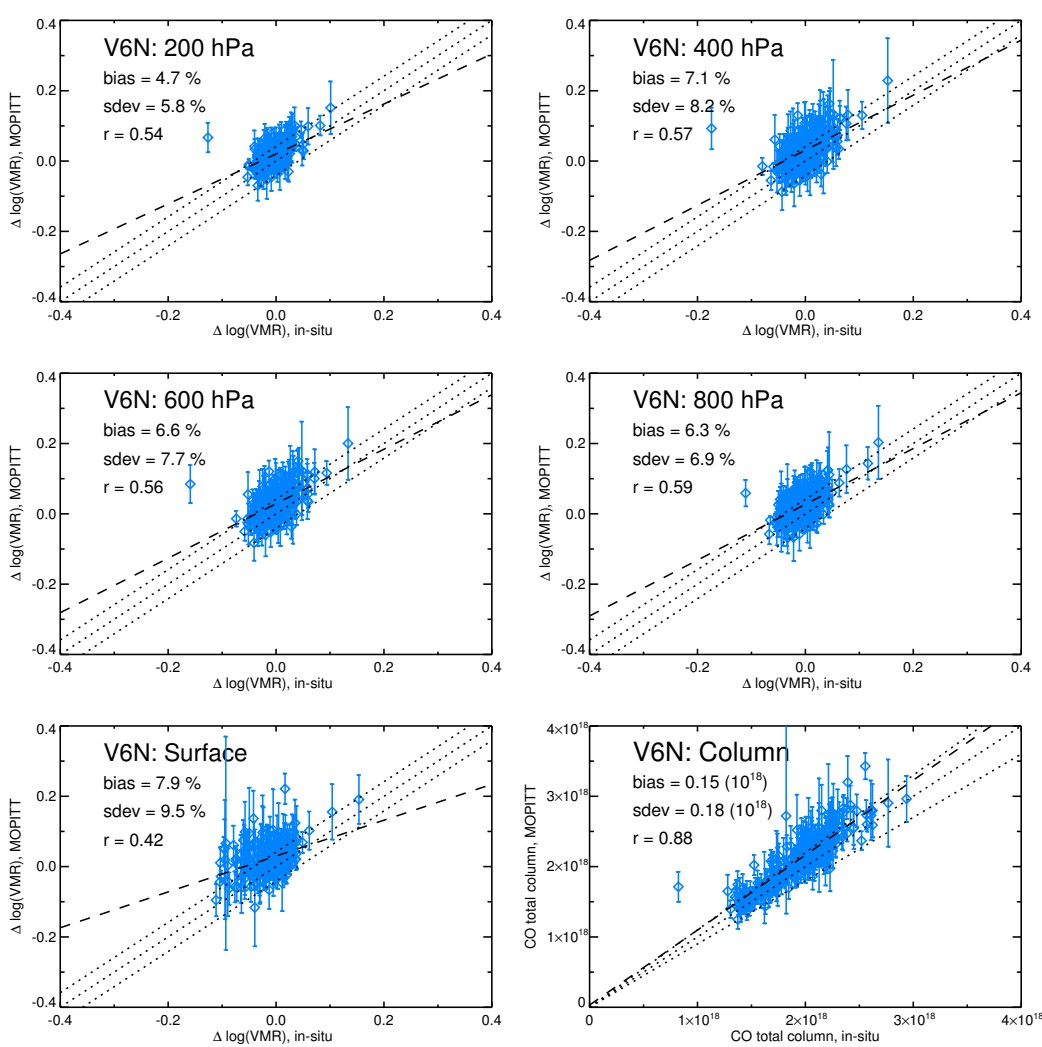

**Fig. 5.** V6 NIR-only validation results based on the NOAA flask measurements. See caption to Figure 3.

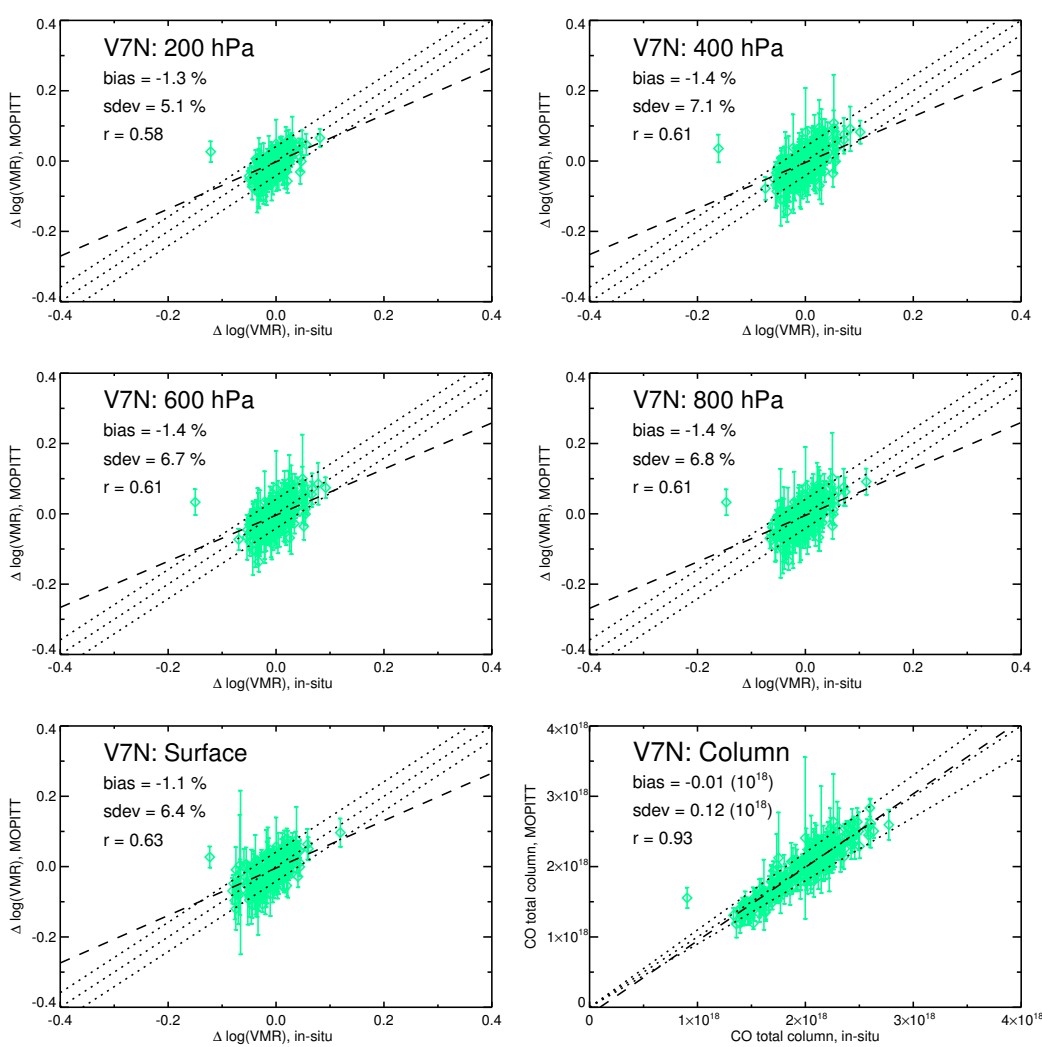

**Fig. 6.** V7 NIR-only validation results based on the NOAA flask measurements. See caption to Figure 3.

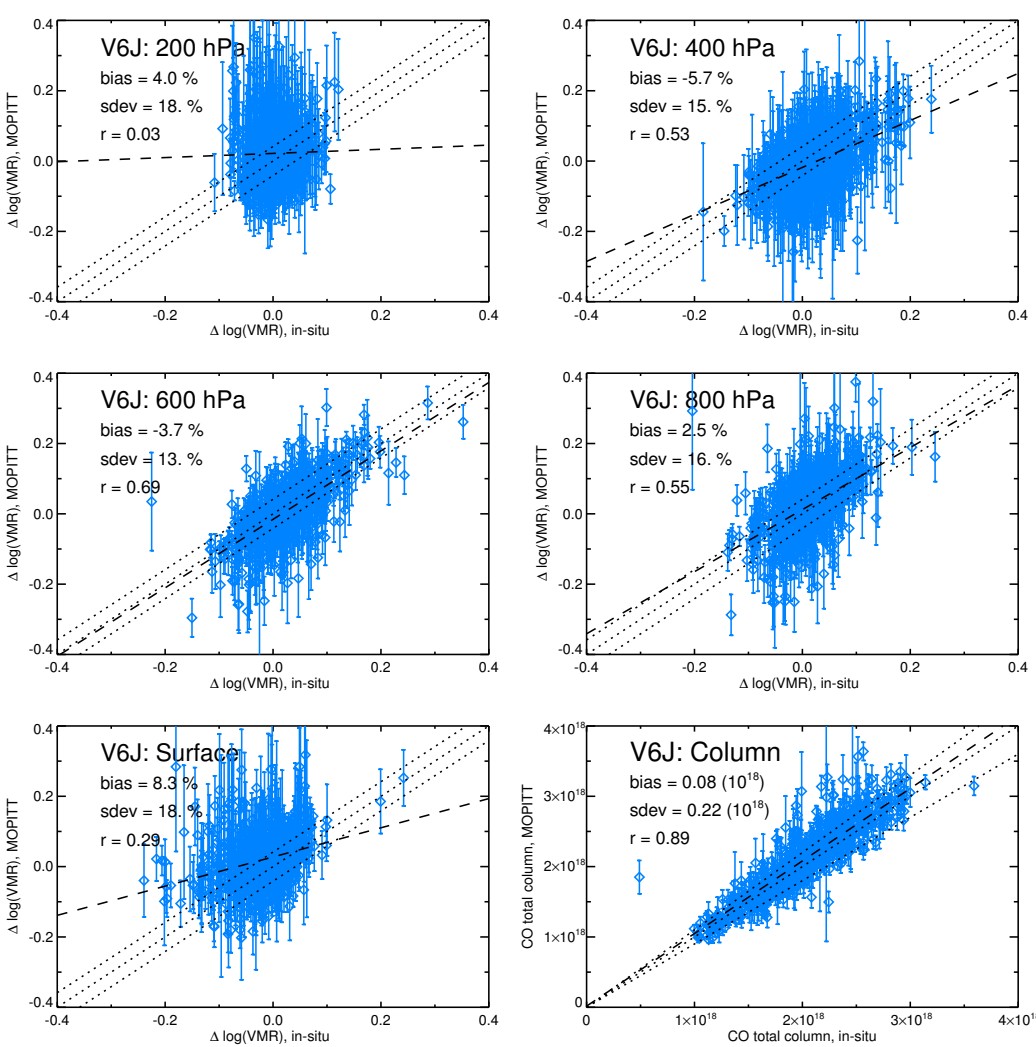

**Fig. 7.** V6 TIR-NIR validation results based on the NOAA flask measurements. See caption to Figure 3.

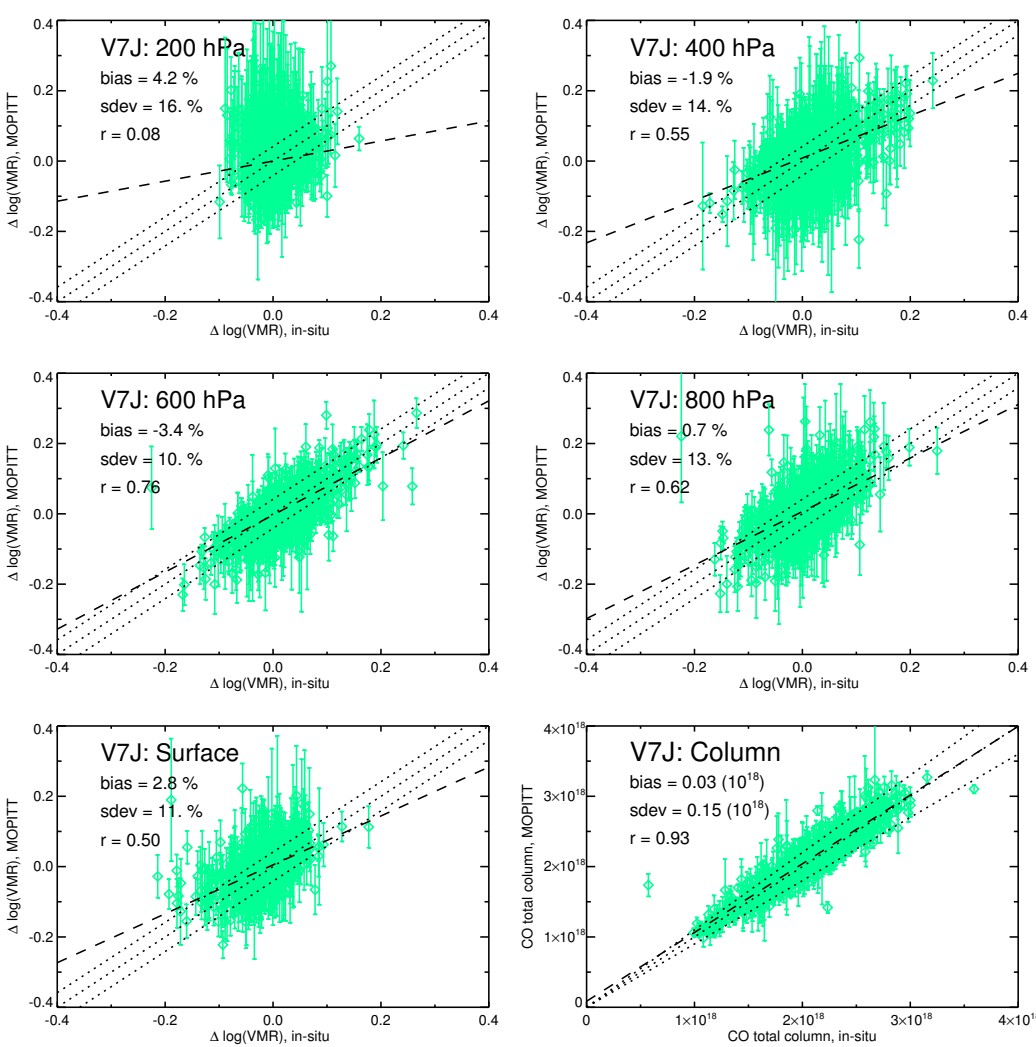

**Fig. 8.** V7 TIR-NIR validation results based on the NOAA flask measurements. See caption to Figure 3.

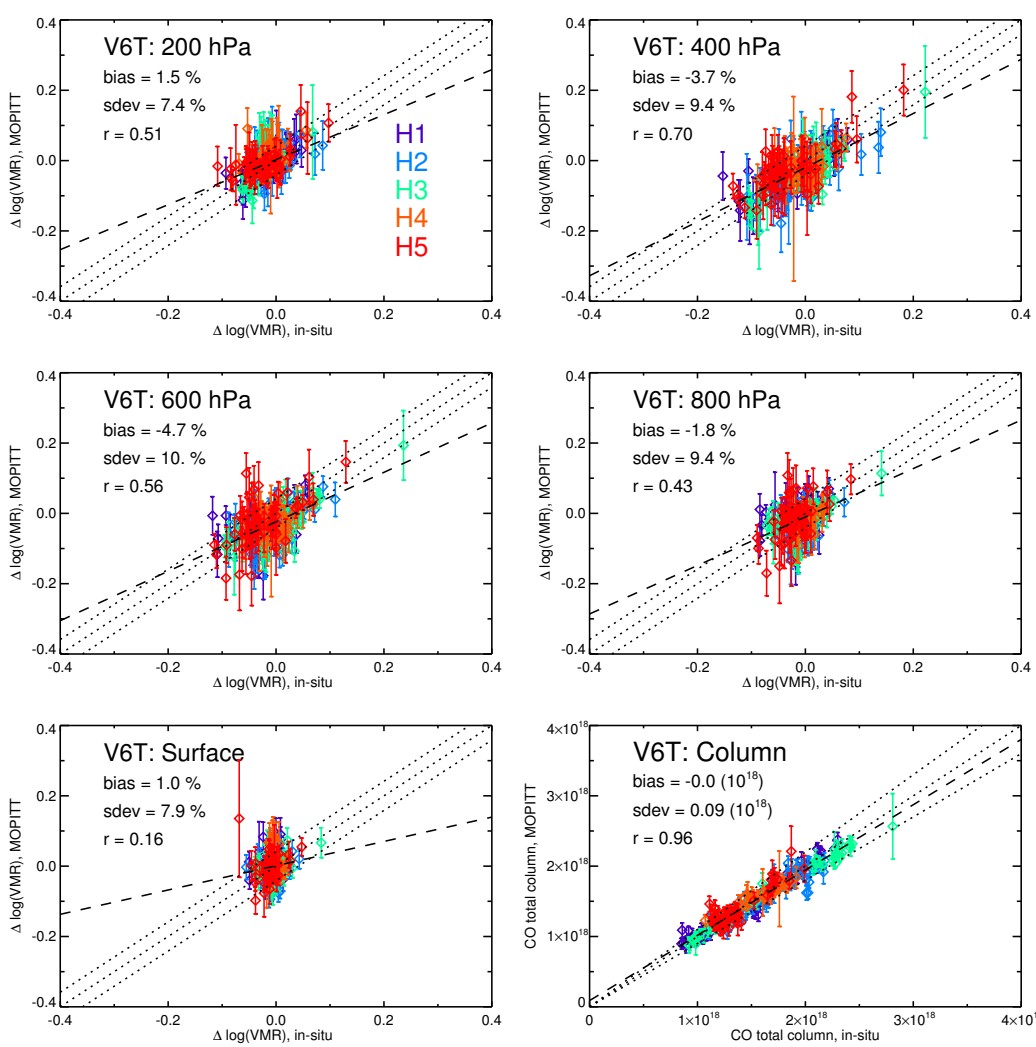

**Fig. 9.** V6 TIR-only validation results based on the HIPPO in-situ profiles. Results for Phase 1 (H1), Phase 2 (H2), etc., are identified by color. See caption to Figure 3.

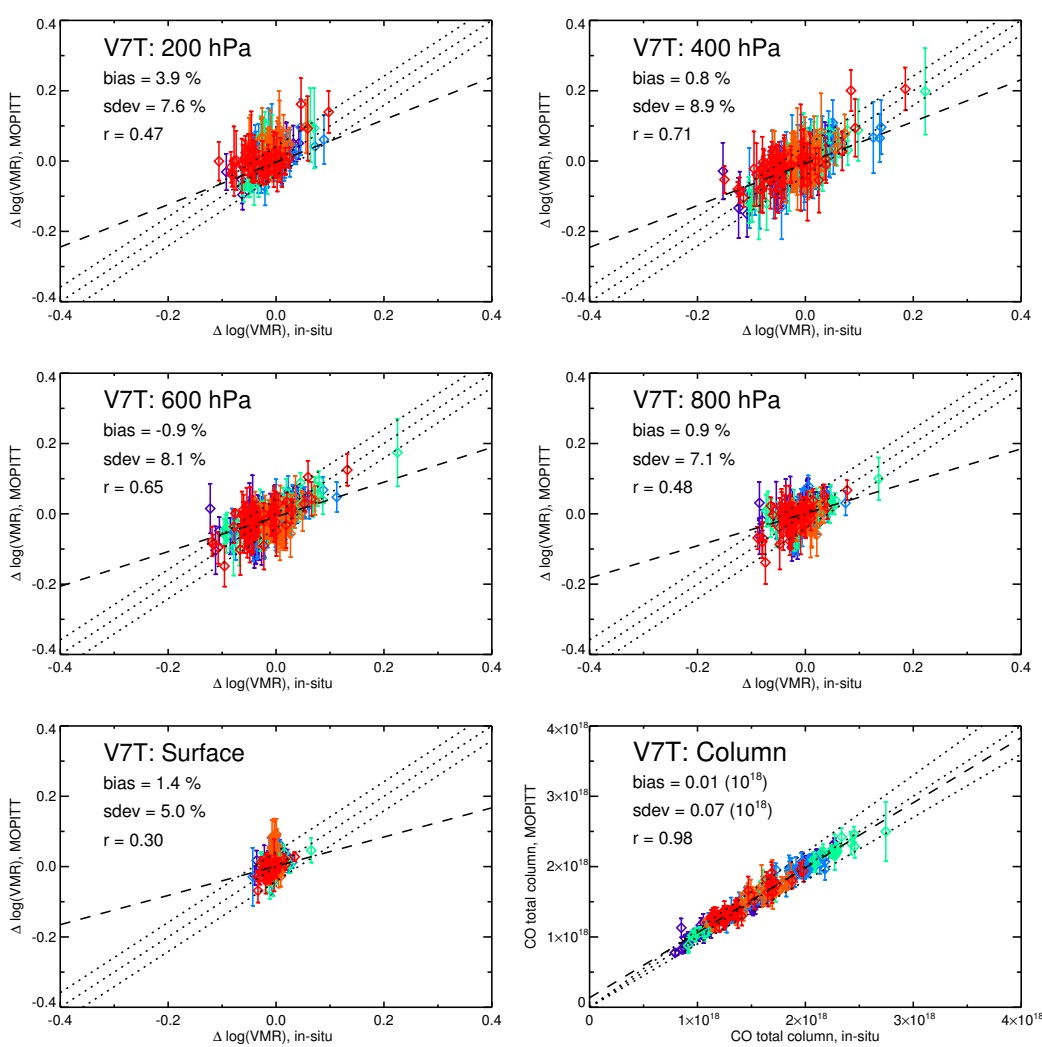

**Fig. 10.** V7 TIR-only validation results based on the HIPPO in-situ profiles. See caption to Figure 3.

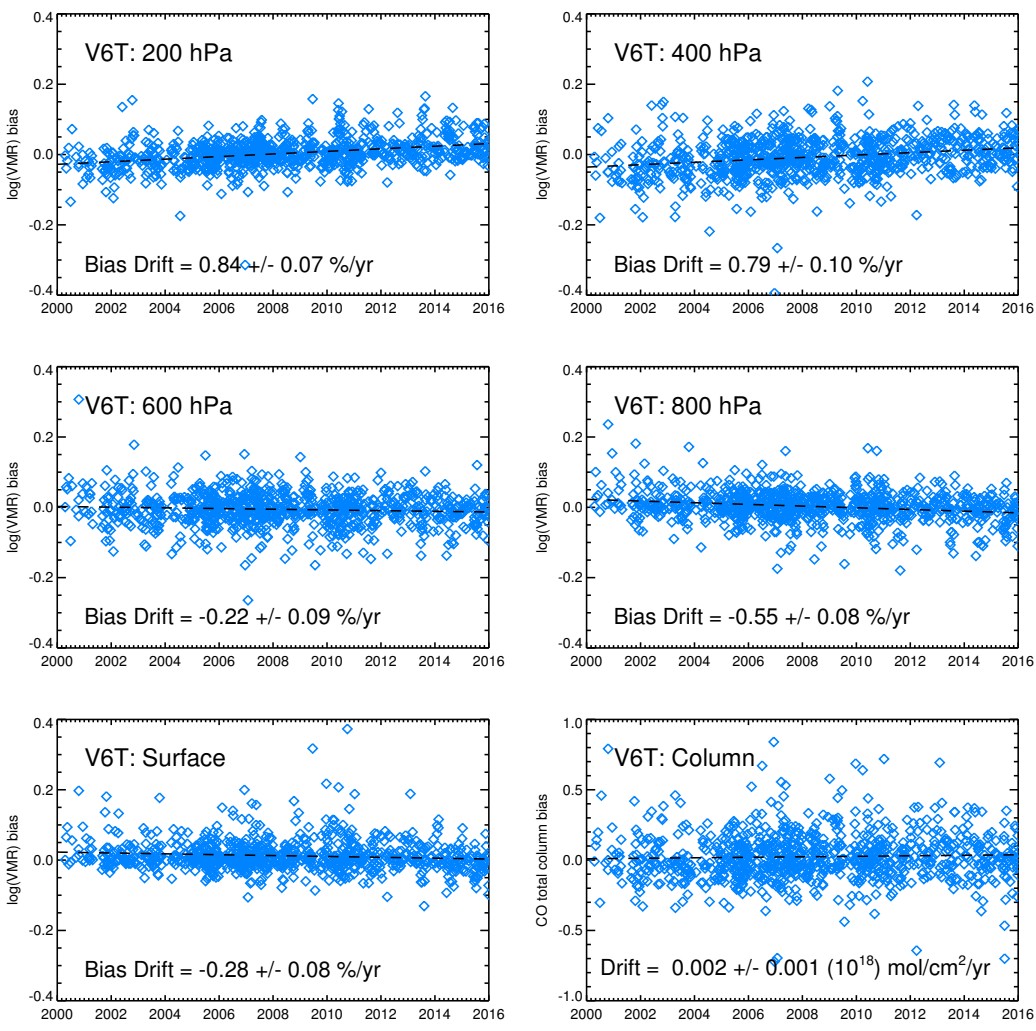

**Fig. 11.** Retrieval bias drift for V6 TIR-only products based on the NOAA flask measurements. CO total column bias values are given in units of $10^{18}$ molec/cm$^2$.

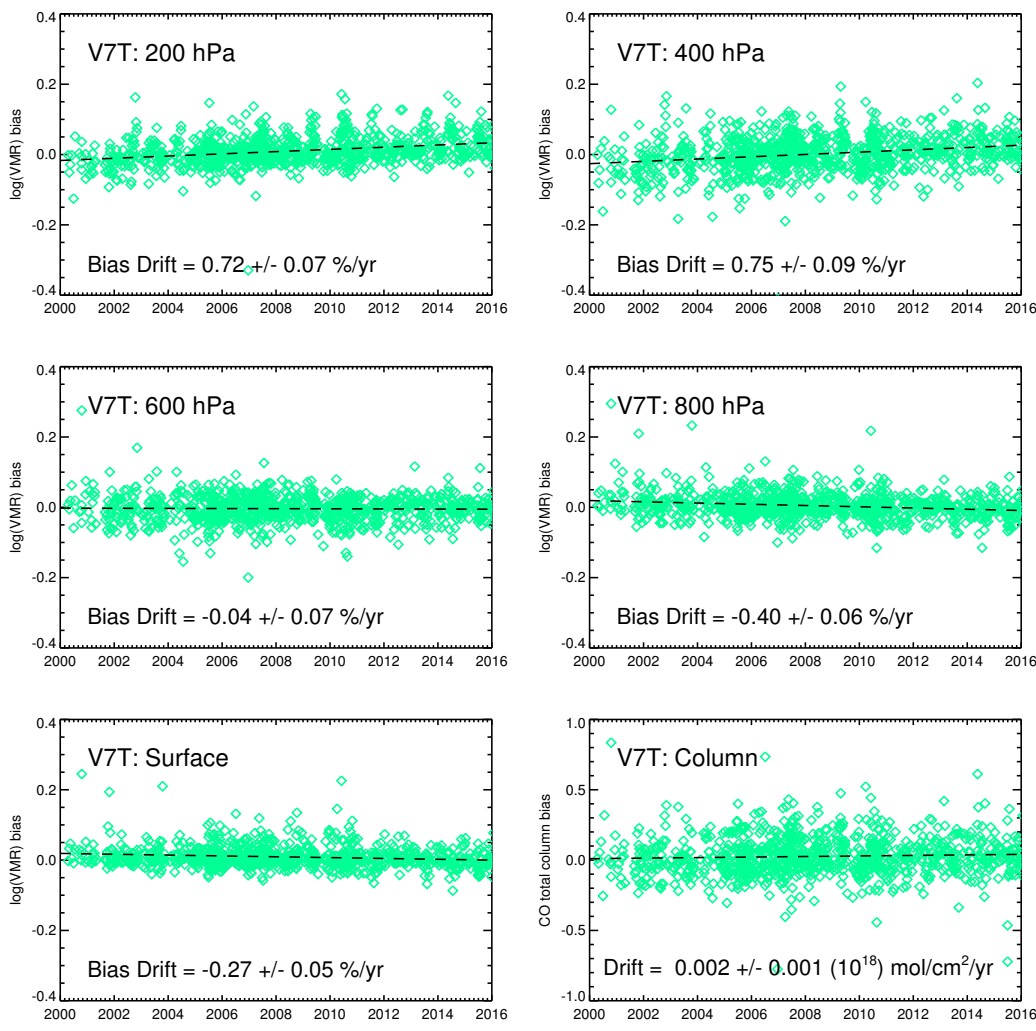

**Fig. 12.** Retrieval bias drift for V7 TIR-only products based on the NOAA flask measurements. See caption to Figure 11.

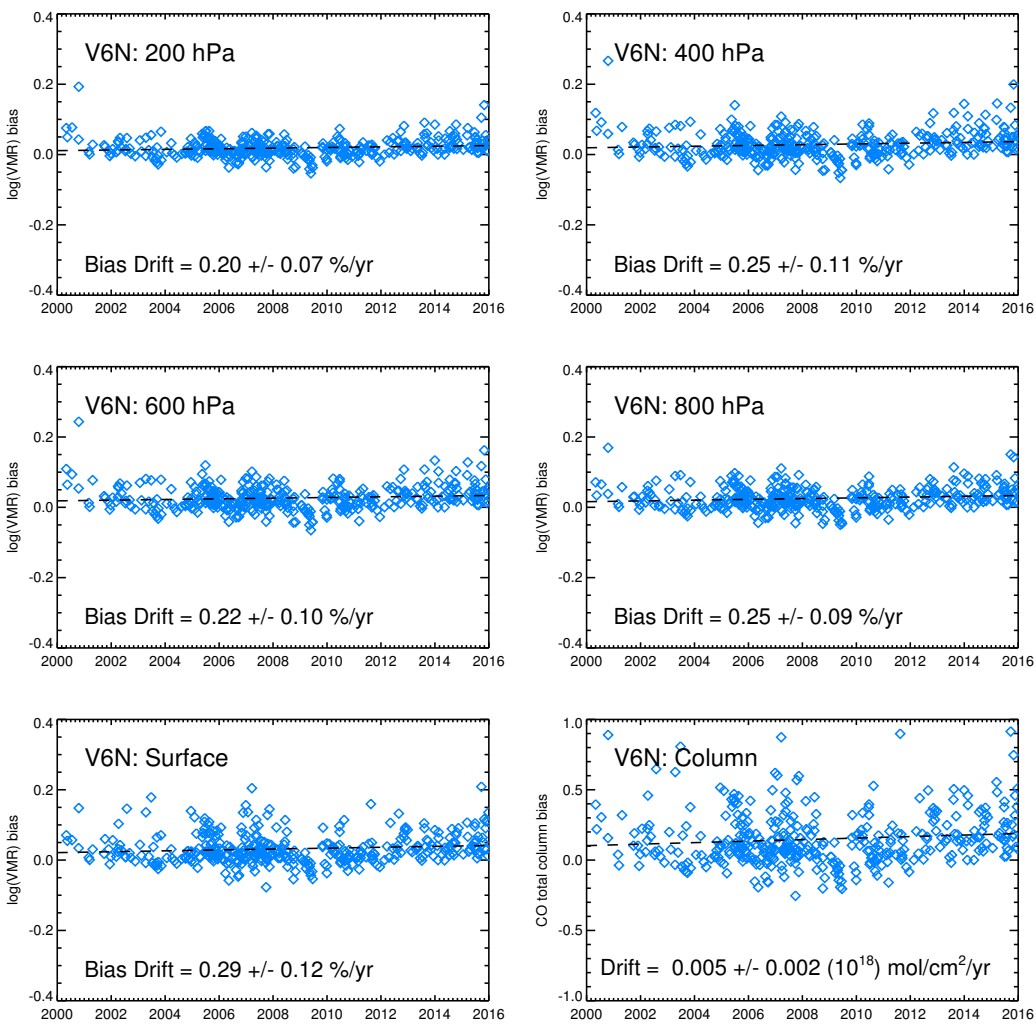

**Fig. 13.** Retrieval bias drift for V6 NIR-only products based on the NOAA flask measurements. See caption to Figure 11.

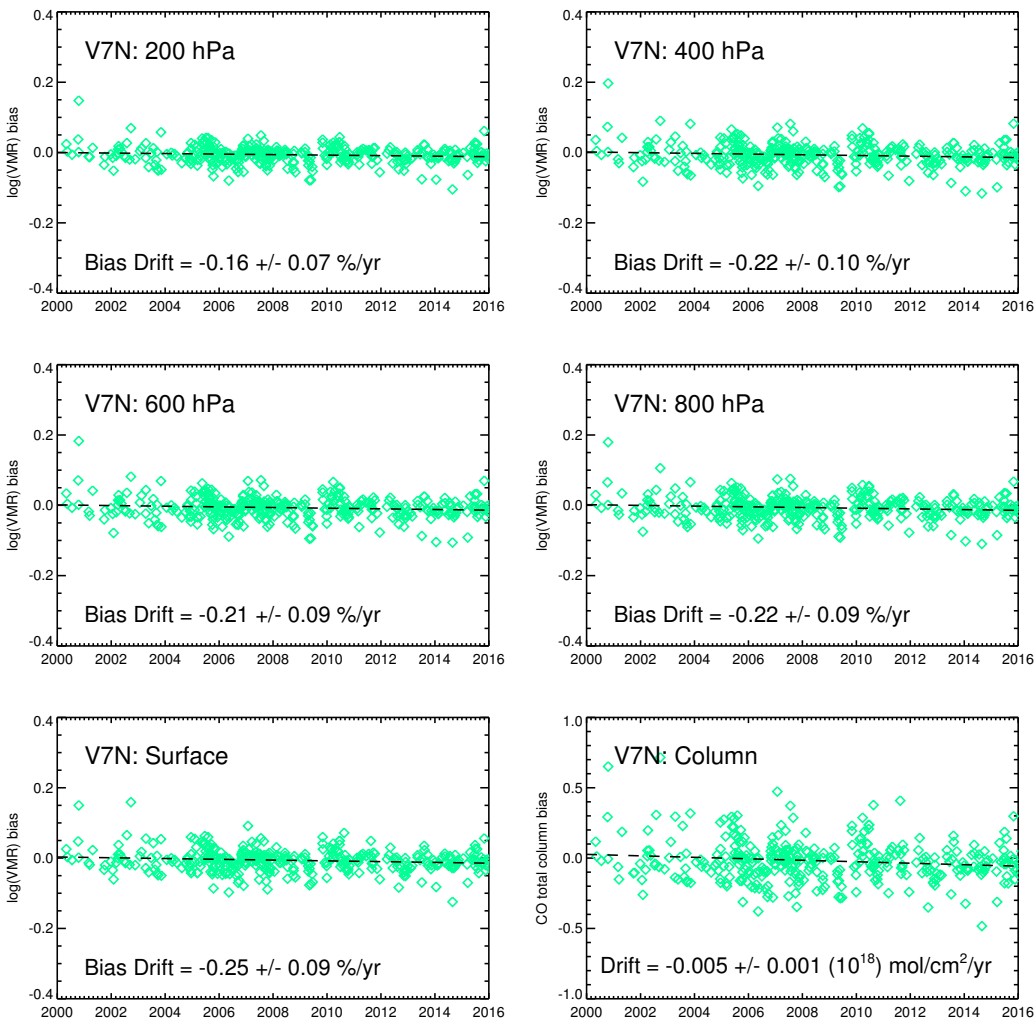

**Fig. 14.** Retrieval bias drift for V7 NIR-only products based on the NOAA flask measurements. See caption to Figure 11.

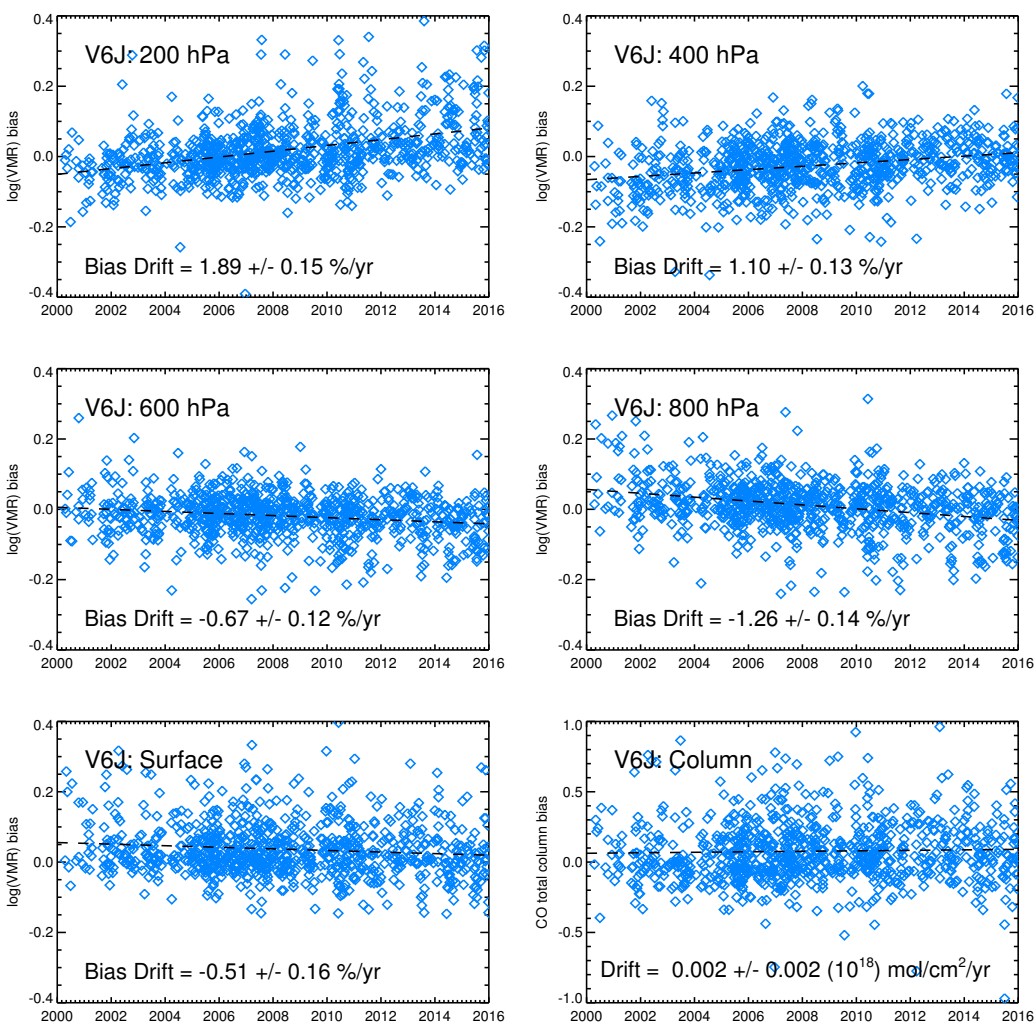

**Fig. 15.** Retrieval bias drift for V6 TIR-NIR products based on the NOAA flask measurements. See caption to Figure 11.

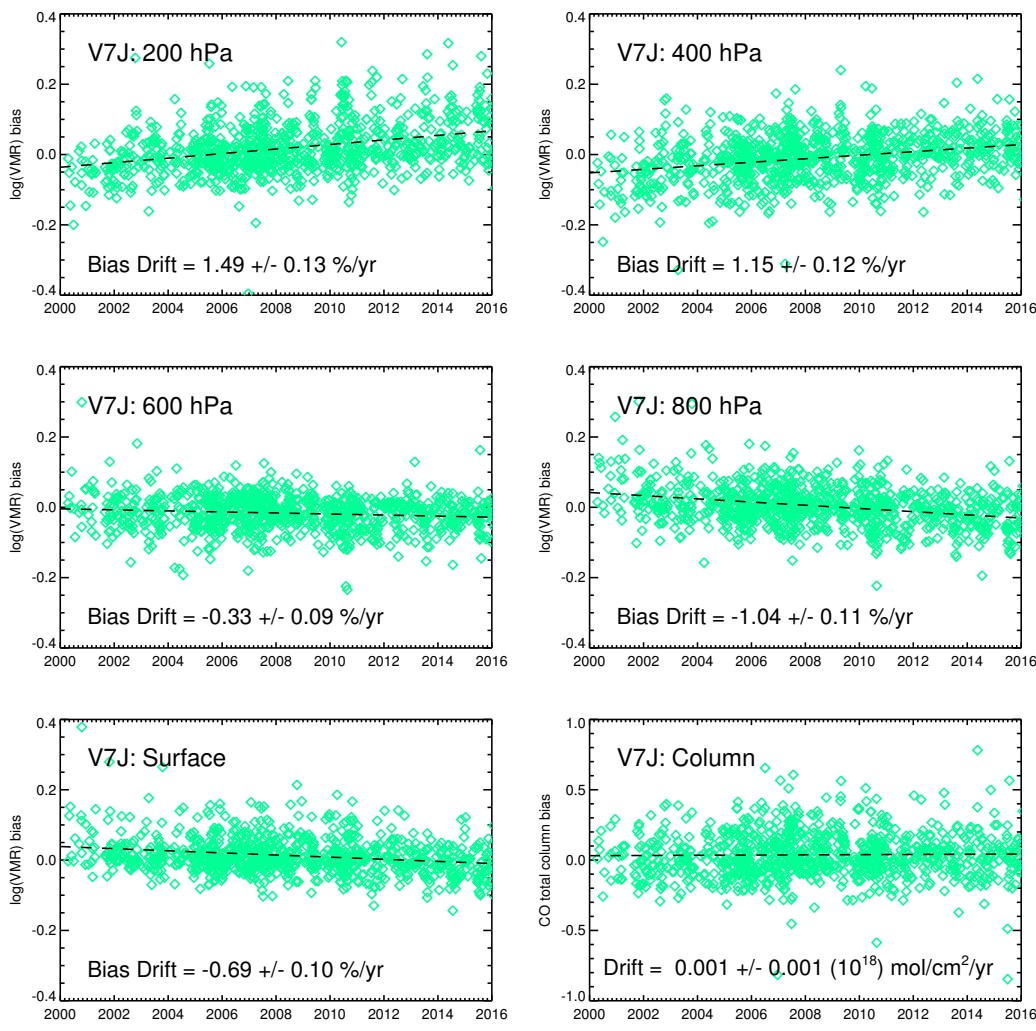

**Fig. 16.** Retrieval bias drift for V7 TIR-NIR products based on the NOAA flask measurements. See caption to Figure 11.

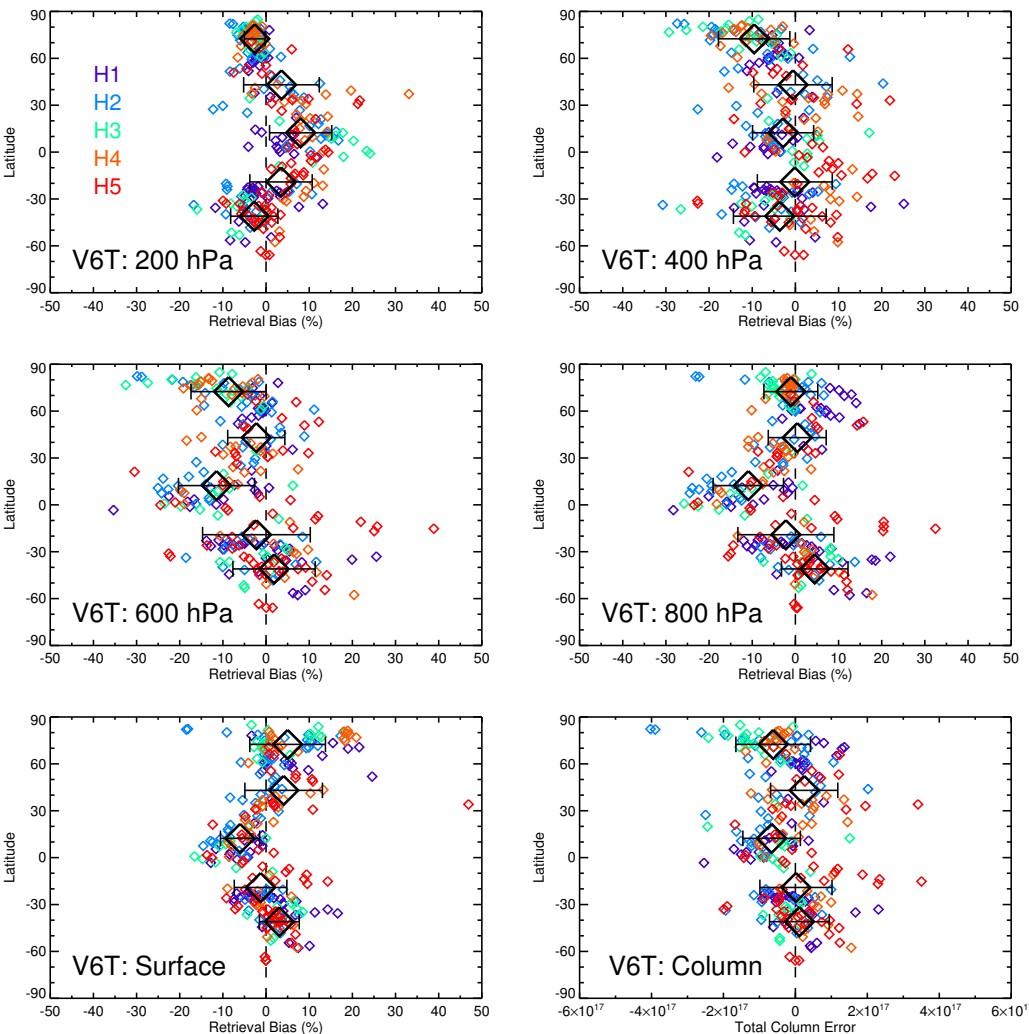

**Fig. 17.** Latitude dependence of V6 TIR-only biases based on the HIPPO CO profiles. Large black diamonds and error bars in each panel indicate bias statistics (mean and standard deviation) representing each 30 degree-wide latitudinal zone. Total column error is given in units of molec/cm$^2$.

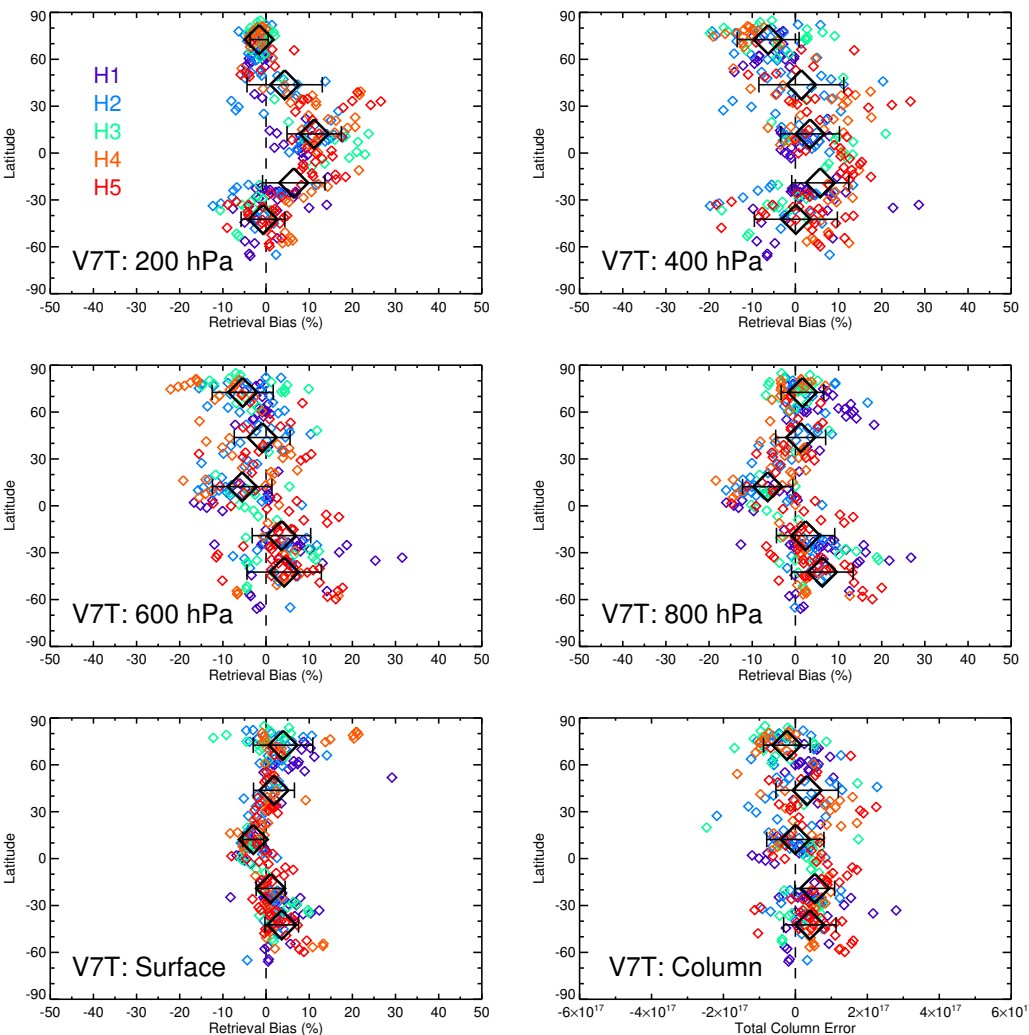

**Fig. 18.** Latitude dependence of V7 TIR-only biases based on the HIPPO CO profiles. See caption to Figure 17.

**Table 1.** Summarized validation results for V6 and V7 TIR-only (V6T and V7T), NIR-only (V6N and V7N) and TIR-NIR (V6J and V7J) products based on in-situ data from NOAA validation sites. Bias and standard deviation statistics for the total column are in units of $10^{18}$ ~~mol~~molec/cm$^2$. Bias and standard deviations for retrieval levels are expressed in %. Correlation coefficients (r) for profile levels are based on differences between retrieved quantities and corresponding a priori quantities, as described in Section 3.1. Total column drift values are provided both in units of $10^{18}$ ~~mol~~molec/cm$^2$/yr and %/yr (in parentheses). Drift for the retrieval levels is expressed in %/yr.

| | | Total Column | Surface | 800hPa | 600hPa | 400hPa | 200hPa |
|---|---|---|---|---|---|---|---|
| V6T | bias | 0.03 | 3.0 | 0.7 | -0.8 | -1.2 | 0.8 |
| | sdev | 0.17 | 9.9 | 9.9 | 10.0 | 12.0 | 9.4 |
| | r | 0.93 | 0.49 | 0.66 | 0.73 | 0.64 | 0.33 |
| | drift | $0.002 \pm 0.001\ (0.07 \pm 0.08)$ | $-0.28 \pm 0.08$ | $-0.55 \pm 0.08$ | $-0.22 \pm 0.09$ | $0.79 \pm 0.10$ | $0.84 \pm 0.07$ |
| V7T | bias | 0.03 | 2.0 | 1.0 | -0.6 | 0.6 | 2.3 |
| | sdev | 0.13 | 6.9 | 8.0 | 8.4 | 11.0 | 9.0 |
| | r | 0.95 | 0.62 | 0.73 | 0.78 | 0.66 | 0.35 |
| | drift | $0.002 \pm 0.001\ (0.07 \pm 0.07)$ | $-0.27 \pm 0.05$ | $-0.40 \pm 0.06$ | $-0.04 \pm 0.07$ | $0.75 \pm 0.09$ | $0.72 \pm 0.07$ |
| V6N | bias | 0.15 | 7.9 | 6.3 | 6.6 | 7.1 | 4.7 |
| | sdev | 0.18 | 9.5 | 6.9 | 7.7 | 8.2 | 5.8 |
| | r | 0.88 | 0.42 | 0.59 | 0.56 | 0.57 | 0.54 |
| | drift | $0.005 \pm 0.002\ (0.32 \pm 0.24)$ | $0.29 \pm 0.12$ | $0.25 \pm 0.09$ | $0.22 \pm 0.10$ | $0.25 \pm 0.11$ | $0.20 \pm 0.07$ |
| V7N | bias | -0.01 | -1.1 | -1.4 | -1.4 | -1.4 | -1.3 |
| | sdev | 0.12 | 6.4 | 6.8 | 6.7 | 7.1 | 5.1 |
| | r | 0.93 | 0.63 | 0.61 | 0.61 | 0.61 | 0.58 |
| | drift | $-0.005 \pm 0.001\ (-0.31 \pm 0.12)$ | $-0.25 \pm 0.09$ | $-0.22 \pm 0.09$ | $-0.21 \pm 0.09$ | $-0.22 \pm 0.10$ | $-0.16 \pm 0.17$ |
| V6J | bias | 0.08 | 8.3 | 2.5 | -3.7 | -5.7 | 4.0 |
| | sdev | 0.22 | 18.0 | 16.0 | 13.0 | 15.0 | 18.0 |
| | r | 0.89 | 0.29 | 0.55 | 0.69 | 0.53 | 0.03 |
| | drift | $0.002 \pm 0.002\ (-0.10 \pm 0.16)$ | $-0.51 \pm 0.16$ | $-1.26 \pm 0.14$ | $-0.67 \pm 0.12$ | $1.10 \pm 0.13$ | $1.89 \pm 0.15$ |
| V7J | bias | 0.03 | 2.8 | 0.7 | -3.4 | -1.9 | 4.2 |
| | sdev | 0.15 | 11.0 | 13.0 | 10.0 | 14.0 | 16.0 |
| | r | 0.93 | 0.50 | 0.62 | 0.76 | 0.55 | 0.08 |
| | drift | $0.001 \pm 0.001\ (-0.04 \pm 0.10)$ | $-0.69 \pm 0.10$ | $-1.04 \pm 0.11$ | $-0.33 \pm 0.09$ | $1.15 \pm 0.12$ | $1.49 \pm 0.13$ |

**Table 2.** Summarized validation results for V6T and V7T products based on in-situ data from HIPPO field campaign. See caption to Table 1.

|     |      | Total Column | Surface | 800hPa | 600hPa | 400hPa | 200hPa |
|-----|------|--------------|---------|--------|--------|--------|--------|
| V6T | bias | -0.02        | 1.0     | -1.8   | -4.7   | -3.7   | 1.5    |
|     | sdev | 0.09         | 7.9     | 9.4    | 10.0   | 9.4    | 7.4    |
|     | r    | 0.96         | 0.16    | 0.43   | 0.56   | 0.70   | 0.51   |
| V7T | bias | 0.01         | 1.4     | 0.9    | -0.9   | 0.8    | 3.9    |
|     | sdev | 0.07         | 5.0     | 7.1    | 8.1    | 8.9    | 7.6    |
|     | r    | 0.98         | 0.30    | 0.48   | 0.65   | 0.71   | 0.47   |

**Table 3.** Latitude dependence of validation results for V6T and V7T products based on in-situ data from HIPPO field campaign. See caption to Table 1.

|     |         |      | Total Column | Surface | 800hPa | 600hPa | 400hPa | 200hPa |
|-----|---------|------|--------------|---------|--------|--------|--------|--------|
| V6T | 60N:90N | bias | -0.06        | 5.0     | -1.0   | -8.7   | -9.5   | -2.6   |
|     |         | sdev | 0.10         | 8.8     | 6.2    | 8.7    | 8.3    | 2.4    |
|     | 30N:60N | bias | 0.02         | 4.1     | 0.4    | -2.3   | -0.6   | 3.6    |
|     |         | sdev | 0.09         | 9.0     | 6.7    | 6.6    | 9.1    | 8.8    |
|     | Eq:30N  | bias | -0.06        | -6.0    | -10.9  | -11.5  | -2.9   | 8.0    |
|     |         | sdev | 0.08         | 4.5     | 8.2    | 8.8    | 7.1    | 7.2    |
|     | 30S:Eq  | bias | 0.00         | -1.3    | -2.2   | -2.2   | -0.1   | 3.5    |
|     |         | sdev | 0.10         | 6.1     | 11.1   | 12.5   | 8.7    | 7.2    |
|     | 60S:30S | bias | 0.01         | 3.1     | 4.5    | 1.9    | -3.6   | -2.7   |
|     |         | sdev | 0.08         | 4.6     | 7.7    | 9.5    | 10.7   | 5.5    |
| V7T | 60N:90N | bias | -0.02        | 3.9     | 1.6    | -5.4   | -6.3   | -1.6   |
|     |         | sdev | 0.06         | 6.9     | 4.9    | 7.1    | 7.2    | 2.1    |
|     | 30N:60N | bias | 0.03         | 1.8     | 1.2    | -0.9   | 1.4    | 4.3    |
|     |         | sdev | 0.08         | 4.7     | 5.8    | 6.5    | 9.8    | 8.7    |
|     | Eq:30N  | bias | 0.00         | -3.0    | -6.4   | -5.5   | 3.4    | 11.2   |
|     |         | sdev | 0.08         | 2.5     | 5.8    | 6.9    | 6.8    | 6.3    |
|     | 30S:Eq  | bias | 0.05         | 1.1     | 2.4    | 3.6    | 5.8    | 6.4    |
|     |         | sdev | 0.05         | 3.4     | 6.7    | 6.8    | 6.6    | 7.2    |
|     | 60S:30S | bias | 0.04         | 3.6     | 6.3    | 4.2    | 0.1    | -0.7   |
|     |         | sdev | 0.07         | 3.9     | 7.1    | 8.6    | 9.6    | 5.1    |