# Peer review of "A Climate-scale Satellite Record for Carbon Monoxide: The MOPITT Version 7 Product"

_Atmospheric Measurement Techniques, 2017_

## Referee Comment (RC1) · Anonymous Referee #1 · 25 Apr 2017

This is a nice paper that provides interesting and useful information on the latest version of the widely used, highly regarded, and long (∼17-year) record of CO profile measurements from the spaceborne MOPITT instrument. The work is directly aligned with the AMT journal. A few of the points made could be expanded upon as I describe below, and some minor rewordings and clarifications are also suggested herein. Once these are addressed, I'd be very happy to recommend the paper for publication in AMT. The standard of writing and graphics etc. is very good.

The main area where I feel this paper could include more information is a greater analysis and breakdown of the relative contribution of each V6/V7 algorithm/software modification to the changes seen in the results. For example, it would be nice to have some quantification of the degree to which the inclusion of time-varying N2O in the retrieval affects the trends shown in figures 10-15. Given the discussion at the end of

section 2.1, this specific example should be easily answerable (at least approximately) with information the team have on hand. In cases, like this, where the team already have sufficient information to make such assessments (e.g., from some test run during development that only changed one of the factors), I'd encourage them to include them in the discussion. I'm not pushing for additional test runs here (though if they'd be easy enough, I guess there's no harm in doing them).

Also, I feel that, in many cases, the discussion would benefit from inclusion of an estimate of the statistical significance of the correlations found (I think it's the t-test for the correlation coefficient isn't it? Though some cases might need autocorrelation to be factored in).

==== More specific comments

Section 1: I feel just a little more background information on CO would be good. Just a few sentences talking about sources and sinks, the role of CO in air quality, use as a marker of more complex pollutants etc. As it is the introduction leaps fairly rapidly into MOPITT specifics. I recognize this is AMT and not, for example, ACP. However, such information never hurts.

Line 13: Are the air quality forecasts mentioned here operational ones (e.g., EPA? Europe?)? In any case, a citation would be good.

Section 2.3: I felt some of this discussion was a little unclear to those (like me) uninitiated into the details of the MOPITT products. For example, on line 93, you use the term "discarded". Does this mean that such profiles get labeled with one of the diagnostic index values (1-5) that means something like "We didn't attempt a retrieval for this profile", or is the profile simply absent from the record entirely? Now they are "retained", what does that mean exactly? Are they now given a different index value (perhaps that's what's described in the sentences that follow, but it's not clear)? Then, on line 99, how is "clear sky" defined? I presume from the thermal channel radiances, but is it also/instead reflected in or given by the diagnostic index values? It feels odd to

specifically define index value 6, but not tell us what values 1 to 5 mean (a table would be good here). Fundamentally, something more in the way of a "MOPITT screening rules 101" discussion might be good to fold in here.

Line 107/108: I don't think you need to enumerate (1) and (2) here as you don't refer to them again. Actually, they initially confused me as I was expecting each of (1) and (2) to contain a comparison, rather than the comparison being of (1) and (2).

Line 113: A bit more detail on the minimizing here might be good (though perhaps it's in the papers cited). Not how the minimization was performed specifically, more about how shallow the minimum was? Was there a clear answer or was (as I suspect) there just a rather incremental improvement in some chi-squared like statistic. This kind of goes back to my main point above related to how big a contribution each change made.

Line 120: "the NOAA aircraft profile set". This dataset hasn't been properly introduced yet, so "a NOAA..." (or something a little more verbose) might be a better introduction.

Line 135: Perhaps this more than a minor point, and clearly goes beyond the scope of this paper. If factoring in the "future" calibrations makes a 20% difference, then to me MOPITT needs to spend more time doing the hot calibrations. 20% is a big number! Are the calibration changes monotonic? How confident are you that the current (14 month?) calibration cycle is frequent enough? Do you have any insights into stability on shorter timescales? I recognize the value, particularly for a sustained record of this duration, of changing observation patterns as little as possible, but I come back to 20% being big.

Lines 200-205: I heartily approve of this approach of looking for correlations in differences from a priori rather than in the "raw" measurements. However, this discussion feels like it's in the wrong place. It applies not only to the NOAA comparisons in this section, but to all the comparisons doesn't it? Perhaps move it into the parent section.

Line 200: There is potential confusion for the uninitiated reader in that the term x_sim

does not appear in equation 1. Might be good to clarify.

Lines 243-249: I think this discussion would be clearer if you swapped the order/sense of the second sentence. "However, because of the sparseness of aircraft in-situ measurements at high altitudes (e.g., pressures less than 350 hPa), particularly for the NOAA dataset, statistical comparisons of V6 and V7 upper-tropospheric CO products are less significant than comparisons of results for the lower troposphere. For example, for retrievals of CO at 200 hPa, the sections of the NOAA validation profiles in the upper troposphere and lower stratosphere are heavily based on the CAM-chem climatology (as described in Section 3.1), and validation results will likely be less reliable than for lower levels."

Line 253: I think it would be better to write this specifically along the lines of "Figures 2 and 3 show ...". Otherwise those figures are not introduced anywhere. You mention them in passing on lines 196 and 201, but with a "don't look at the figures now, just bear these points in mind when it's time to do so" message. Without a formal introduction here (line 253), the reader may think that this mention is like those others, rather than an indication that it's time to pay attention to them.

Line 257: I'm not sure "substantially" is justified here. These changes are small compared to the -4.7% to -0.9% improvement you've just told us about. Perhaps soften the tone.

Line 296: Perhaps reminder the reader of exactly how long it is, add "(17 years to date)" or something like that.

Lines 329-332: Back to my main point at the top, if you have to hand test runs that quantify the relative contributions of these terms, it would be good to summarize them.

Captions for Figures 3-9: Reword as "As for Figure 2 except ...", that way you get quasi-complete caption that includes description of the dashed lines etc.

Caption for Figure 16: Would be good to define black diamond here in the caption as

well as in body text.

Caption for Figure 17: Either include definition of black diamond here too, or use the same "As figure Figure 16 except..." construct advocated previously. (Don't see the need to advocate similar for Figures 11-15 as their captions are complete as they are).

Table 2 caption: Infinite loop reference to Table 2 caption. Did you mean Table 1?

Table 3 caption: Presumably should point to Table 1 caption here too?
* * *

---

## Referee Comment (RC2) · Anonymous Referee #2 · 4 May 2017

This paper aims at presenting the features of the MOPITT V7 CO data and validation results. Statistical comparisons are performed using aircraft in-situ measurements as the reference. Improvements are demonstrated for V7 products in comparison with V6 products. The paper is very well written, as the previous MOPITT papers by Deeter et al. In contrast to the previous MOPITT validation papers, the statistics are performed in terms of difference between the retrieved and a priori VMR values (except for the total column). This choice is wise and well detailed in Section 3.1. The subject of the paper is appropriate to AMT. I found the paper clearly presented, well organized and essential for the wide community of the MOPITT data users. I recommend the paper to be published, after the few minor issues I raise below are addressed.

Line 97: "V5 User's Guide" Add a reference?

[Figure]

Line 161: it is clearly said that "in-situ measurements are assumed to be exact". Would you add somewhere how the bias in percentage is calculated? (MOPITT – in situ / in situ?)

Line 278: "The bias in total column is reduced from 0.16" According to Figure 6 and Table 1, it should be 0.15 not 0.16.

It should be nice to have a table online where the different features of all the MOPITT data versions would be summarized.
* * *

---

## Author Comment (AC1) · 23 May 2017

Original reviewers' comments in blue.  Authors' responses in black. **Manuscript revisions noted in bold.**

**Replies to Comments of Reviewer #1**

General Comments:

1. The main area where I feel this paper could include more information is a greater analysis and breakdown of the relative contribution of each V6/V7 algorithm/software modification to the changes seen in the results. For example, it would be nice to have some quantification of the degree to which the inclusion of time-varying N2O in the retrieval affects the trends shown in figures 10-15. Given the discussion at the end of section 2.1, this specific example should be easily answerable (at least approximately) with information the team have on hand. In cases, like this, where the team already have sufficient information to make such assessments (e.g., from some test run during development that only changed one of the factors), I'd encourage them to include them in the discussion. I'm not pushing for additional test runs here (though if they'd be easy enough, I guess there's no harm in doing them).

Authors' Reply: The authors would like to thank Reviewer #1 for providing a thorough review of our paper and offering many useful suggestions.  Tests to quantify the sensitivity of the validation results to individual changes in the retrieval algorithm would not be useful in a cumulative sense because of the strong coupling of these effects in radiative transfer.  Thus, testing each improvement individually would not necessarily lead to an understanding of the overall bias drift.  For example, retrieval biases due to N2O involve thermal contrast effects, which depend also on the specified temperature profile and the radiance bias correction factors.  N2O bias sensitivity tests will therefore generally produce different results depending on whether MERRA or MERRA-2 reanalyses are used for the tests and will depend also on the chosen radiance bias factors.  On the other hand, in the case of N2O growth, we did perform simulations during the development of V7 to investigate associated retrieval biases.  **These results might be of interest to some users and have therefore been added to Section 2.1, along with a new figure.**

2. Also, I feel that, in many cases, the discussion would benefit from inclusion of an estimate of the statistical significance of the correlations found (I think it's the t-test for the correlation coefficient isn't it? Though some cases might need autocorrelation to be factored in).

Authors' Reply: **Following the reviewer's suggestion, we have analyzed the statistical significance of both the scatterplots presented in Sections 3.1 and 3.2 and the bias drift results presented in Section 4.4 and have added text describing the results.**

Specific Comments:

1. Section 1: I feel just a little more background information on CO would be good. Just a few sentences talking about sources and sinks, the role of CO in air quality, use as a marker of more complex pollutants etc. As it is the introduction leaps fairly rapidly into MOPITT specifics. I recognize this is AMT and not, for example, ACP. However, such

information never hurts.

Authors' Reply: **We agree that such information is useful, and have therefore added several sentences to the beginning of Section 1.**

2. Line 13: Are the air quality forecasts mentioned here operational ones (e.g., EPA? Europe?)? In any case, a citation would be good.

Authors' Reply: We refer specifically to operational ECMWF forecasts; **a citation to Inness et al. has been added.**

3. Section 2.3: I felt some of this discussion was a little unclear to those (like me) uninitiated into the details of the MOPITT products. For example, on line 93, you use the term "discarded". Does this mean that such profiles get labeled with one of the diagnostic index values (1-5) that means something like "We didn't attempt a retrieval for this profile", or is the profile simply absent from the record entirely? Now they are "retained", what does that mean exactly? Are they now given a different index value (perhaps that's what's described in the sentences that follow, but it's not clear)? Then, on line 99, how is "clear sky" defined? I presume from the thermal channel radiances, but is it also/instead reflected in or given by the diagnostic index values? It feels odd to specifically define index value 6, but not tell us what values 1 to 5 mean (a table would be good here). Fundamentally, something more in the way of a "MOPITT screening rules 101" discussion might be good to fold in here.

Authors' Reply: As the first sentence of this section describes, only MOPITT observations of clear-sky scenes are passed to the retrieval algorithm. The terms 'discarded' and 'retained' refer to the fate of individual MOPITT observations at the end of the cloud detection algorithm; these are the only two possibilities. Observations which are retained are treated as clear-sky observations by the retrieval algorithm and typically lead to a retrieved profile. Cloud-affected observations are not processed further and therefore are absent from MOPITT retrieval products. Cloud detection diagnostic values are provided for the clear-sky observations so that interested users can understand the basis for finding the observation to be clear. **In response to the Reviewer's comment, we have added a reference to the V5 User's Guide, where cloud detection and diagnostics are discussed in more detail.** Because such details interest only a relatively small group of MOPITT users, we feel that the User's Guides are the best location for this material.

4. Line 107/108: I don't think you need to enumerate (1) and (2) here as you don't refer to them again. Actually, they initially confused me as I was expecting each of (1) and (2) to contain a comparison, rather than the comparison being of (1) and (2).

Authors' Reply: **The two parts of this sentence are no longer enumerated.**

5. Line 113: A bit more detail on the minimizing here might be good (though perhaps it's in the papers cited). Not how the minimization was performed specifically, more about how shallow the minimum was? Was there a clear answer or was (as I suspect) there just a rather incremental improvement in some chi-squared like statistic. This kind of goes back to my main point above related to how big a contribution each change made.

Authors' Reply: The success of the method is demonstrated by the fact that, after optimization, the

overall retrieval biases (based on the HIPPO profiles) at 400 and 800 hPa were both reduced to less than 1%. The potential value of additional details regarding the minimization results to the readers is unclear.

6. Line 120: "the NOAA aircraft profile set". This dataset hasn't been properly introduced yet, so "a NOAA..." (or something a little more verbose) might be a better introduction.

Authors' Reply: **The end of the sentence in question now reads " ... using the NOAA aircraft profile set described in Section 3.1."**

7. Line 135: Perhaps this more than a minor point, and clearly goes beyond the scope of this paper. If factoring in the "future" calibrations makes a 20% difference, then to me MOPITT needs to spend more time doing the hot calibrations. 20% is a big number! Are the calibration changes monotonic? How confident are you that the current (14 month?) calibration cycle is frequent enough? Do you have any insights into stability on shorter timescales? I recognize the value, particularly for a sustained record of this duration, of changing observation patterns as little as possible, but I come back to 20% being big.

Authors' Reply: The 20% value relates only to possible biases in the V7 NIR-only beta product; biases for all of the archival V7 products are expected to be much smaller due to the use of 'bracketing' hot calibrations. Hot calibrations for the MOPITT instrument are performed during a series of instrument operations which are performed approximately annually. No earth-view observations are recorded during this sequence, which typically lasts almost two weeks. Therefore, more frequent hot calibrations would not necessarily improve the quality of V7 archival products but would certainly lead to more frequent gaps in the MOPITT data record.

8. Lines 200-205: I heartily approve of this approach of looking for correlations in differences from a priori rather than in the "raw" measurements. However, this discussion feels like it's in the wrong place. It applies not only to the NOAA comparisons in this section, but to all the comparisons doesn't it? Perhaps move it into the parent section.

Authors' Reply: This is a good point. **The material has been moved to the beginning of Section 3, just before Section 3.1.**

9. Line 200: There is potential confusion for the uninitiated reader in that the term x_sim does not appear in equation 1. Might be good to clarify.

Authors' Reply: Agreed. **The relevant part of this sentence now reads ' ... where MOPITT retrieved VMR values were plotted directly against simulated VMR retrievals x_{sim} (calculated according to the RHS of Eq. 1, with x_{true} based on the in-situ profile), ...'**

10. Lines 243-249: I think this discussion would be clearer if you swapped the order/sense of the second sentence. "However, because of the sparseness of aircraft in-situ measurements at high altitudes (e.g., pressures less than 350 hPa), particularly for the NOAA dataset, statistical comparisons of V6 and V7 upper-tropospheric CO products are less significant than comparisons of results for the lower troposphere. For example, for retrievals of CO at 200 hPa, the sections of the NOAA validation profiles in the upper troposphere and lower stratosphere are heavily based on the CAM-chem climatology

(as described in Section 3.1), and validation results will likely be less reliable than for lower levels."

Authors' Reply: **The text has been revised accordingly.**

11. Line 253: I think it would be better to write this specifically along the lines of "Figures 2 and 3 show ...". Otherwise those figures are not introduced anywhere. You mention them in passing on lines 196 and 201, but with a "don't look at the figures now, just bear these points in mind when it's time to do so" message. Without a formal introduction here (line 253), the reader may think that this mention is like those others, rather than an indication that it's time to pay attention to them.

Authors' Reply: These figures are formally introduced in the first sentence of the second paragraph of Section 3.1.

12. Line 257: I'm not sure "substantially" is justified here. These changes are small compared to the -4.7% to -0.9% improvement you've just told us about. Perhaps soften the tone.

Authors' Reply: **The word 'substantially' has been replaced with 'significantly'.**

13. Line 296: Perhaps reminder the reader of exactly how long it is, add "(17 years to date)" or something like that.

Authors' Reply: **The text has been revised accordingly.**

14. Lines 329-332: Back to my main point at the top, if you have to hand test runs that quantify the relative contributions of these terms, it would be good to summarize them.

Authors' Reply: See authors' reply to first General Comment.

15. Captions for Figures 3-9: Reword as "As for Figure 2 except ...", that way you get quasi-complete caption that includes description of the dashed lines etc.

Authors' Reply: **The sentence 'See caption to Figure 2.' has been added to the captions for these figures.**

16. Caption for Figure 16: Would be good to define black diamond here in the caption as well as in body text.

Authors' Reply: **The caption now describes the meaning of the black diamonds and error bars.**

17. Caption for Figure 17: Either include definition of black diamond here too, or use the same "As figure Figure 16 except..." construct advocated previously. (Don't see the need to advocate similar for Figures 11-15 as their captions are complete as they are).

Authors' Reply: **The caption for Figure 17 has been revised accordingly.**

18. Table 2 caption: Infinite loop reference to Table 2 caption. Did you mean Table 1?

Authors' Reply: Yes.  **This has been corrected.**

19. Table 3 caption: Presumably should point to Table 1 caption here too?

Authors' Reply: **The caption for Table 3 now refers to the Table 1 caption.**

**Replies to Comments of Reviewer 2**

1. Line 97: "V5 User's Guide" Add a reference?

Authors' Reply: **A reference to the V5 User's Guide has been added.**

2. Line 161: it is clearly said that "in-situ measurements are assumed to be exact". Would you add somewhere how the bias in percentage is calculated? (MOPITT – in situ / in situ?)

Authors' Reply: **This is now explicitly described in the second paragraph of Section 3.1.**

3. Line 278: "The bias in total column is reduced from 0.16" According to Figure 6 and Table 1, it should be 0.15 not 0.16.

Authors' Reply: **The text has been corrected.**

4. It should be nice to have a table online where the different features of all the MOPITT data versions would be summarized.

Authors' Reply: We agree and now plan to add such a table on the MOPITT website.

---

## Author Response (AR2)

Original reviewer's comments in blue. Authors' responses in black.

**Replies to Comments of Associate Editor on June 2, 2017**

There are a few minor issues I would like you to resolve for the final version which will be published. Please take into account the following:

Throughout the manuscript text and in the Figures you indicate columns in"mol/cm^2", which means molecules per cm^2. I suggest to abbreviate molecules as molec. and not as mol to avoid confusion with moles.

Authors' Reply: Done.

P1, L14: Besides methane, CO is also formed by oxidation of other volatile organic compounds. Please include that.

Authors' Reply: Done.

P2, L50: please quantify the weak growth in N2O already here (I know you do it later).

Authors' Reply: The purpose of this paragraph is to explain the concept qualitatively. We prefer to just state in this paragraph that N2O concentrations have been gradually increasing (with a relevant reference), and then describe this growth quantitatively slightly later in the text.

P7, L224: please clarify what 'RHS' stands for.

Authors' Reply: 'RHS' has been replaced with 'right-hand side'

Figure 3 and later figures: please indicate in the captions or in the Figures the physical units of the CO total column (for the lower right panels)

Authors' Reply: The last sentence in the caption to Fig. 3 has been revised to 'CO total column values as well as bias and standard deviation statistics for the total column are in units of molec/cm$^2$.' Captions to Figures 4-10 include the sentence 'See caption to Figure 3' and therefore do not require revision. Caption to Figure 11 now includes added sentence 'CO total column bias values are given in units of 10$^{18}$ molec/cm$^2$.' Captions to Figures 12-16 now include added sentence 'See caption to Figure 11.' Caption to Figure 17 now includes added sentence 'Total column error is given in units of molec/cm$^2$.'